# Quantum Entropy Scoring for Fast Robust Mean Estimation and Improved Outlier Detection

**Yihe Dong**
Microsoft Research
yihedong@gmail.com

**Samuel B. Hopkins**
University of California, Berkeley
hopkins@berkeley.edu

**Jerry Li**
Microsoft Research
jerrl@microsoft.com

## Abstract

We study two problems in high-dimensional robust statistics: *robust mean estimation* and *outlier detection*. In robust mean estimation the goal is to estimate the mean $\mu$ of a distribution on $\mathbb{R}^d$ given $n$ independent samples, an $\varepsilon$-fraction of which have been corrupted by a malicious adversary. In outlier detection the goal is to assign an *outlier score* to each element of a data set such that elements more likely to be outliers are assigned higher scores. Our algorithms for both problems are based on a new outlier scoring method we call QUE-scoring based on *quantum entropy regularization*. For robust mean estimation, this yields the first algorithm with optimal error rates and nearly-linear running time $\widetilde{O}(nd)$ in all parameters, improving on the previous fastest running time $\widetilde{O}(\min(nd/\varepsilon^6, nd^2))$. For outlier detection, we evaluate the performance of QUE-scoring via extensive experiments on synthetic and real data, and demonstrate that it often performs better than previously proposed algorithms. Code for these experiments is available at https://github.com/twistedcubic/que-outlier-detection.

## 1 Introduction

We study outlier-robust statistics in high dimensions, focusing on the question: *can theoretically sound outlier robust algorithms have practical running times for large, high-dimensional data sets?* We address two related problems: *robust mean estimation*, which is primarily theoretical, and an applied counterpart, *outlier detection*.

**Robust mean estimation** Our main theoretical contribution is the first nearly-linear time algorithm for robust mean estimation with nearly-optimal error. Here the goal is to estimate the mean $\mu \in \mathbb{R}^d$ of a $d$-dimensional distribution $D$ given $\varepsilon$-*corrupted* samples $X_1, \ldots, X_n$ – that is, i.i.d. samples, an unknown $\varepsilon$-fraction of which have been maliciously corrupted. Under (for instance) the assumption that the covariance of $D$ is bounded by $\mathrm{Id}$, it has been long known to be possible in exponential time to estimate $\mu$ by $\hat{\mu}$ having $\|\mu - \hat{\mu}\|_2 \le O(\sqrt{\varepsilon})$. In particular, this rate of error is independent of $d$.

Polynomial-time algorithms provably achieving such $d$-independent error became known only recently, starting with the works [8, 15]. Until our work, the running time of algorithms with provably $d$-independent error remained suboptimal by polynomial factors in $d$ or $\varepsilon$: the fastest running time achieved before this work was $\widetilde{O}(\min(nd^2, nd/\varepsilon^6))$ [6, 8, 15, 9]. (Here $\widetilde{O}(\cdot)$ notation hides logarithmic factors in $n$ and $d$). While these running times represent a dramatic improvement over previous exponential-time algorithms, there are still many interesting regimes where the additional runtime overheads these algorithms incur render them impractically slow. *We give the first algorithm for robust mean estimation with running time $\widetilde{O}(nd)$ which achieves error $\|\mu - \hat{\mu}\|_2 \le O(\sqrt{\varepsilon})$.* Note that this running time is nearly-linear in the input size $nd$. Similar to prior works, our algorithm has information-theoretically optimal sample complexity and nearly-optimal error rates in both the bounded-covariance and sub-Gaussian regimes.

**Outlier detection** Our main applied contribution is a new algorithm for high-dimensional outlier detection, which we assess via experiments on both synthetic and real data [1]. Our goal is to take a dataset $X_1, \ldots, X_n \in \mathbb{R}^d$ and assign to each $X_i$ an *outlier score* $\tau_i \geq 0$, so that higher scores $\tau_i$ are assigned to points $X_i$ more likely to be outliers. Of course, what constitutes an outlier varies across applications, so no single algorithm for outlier detection is likely to be the best in all domains. We show that our method performs well in settings where individual outliers are difficult to pick out on their own (by, say, their $\ell_2$ norms or their distances to nearby points), but still collectively bias empirical statistics such as the mean and covariance.

We compare our method to baselines based on PCA and Euclidean distances, as well as more sophisticated algorithms from existing literature based on nearest-neighbor distances. Our algorithm has nearly-linear running time in theory, and simple implementations in practice incur minimal overhead beyond standard spectral methods, allowing us to run on 1024-dimensional data with no special optimizations and 8192-dimensional data with a fast approximate implementation. It can therefore be used in practice to complement existing approaches to outlier detection in exploratory data analysis.

## 1.1 What is an outlier and why are they hard to find?

For us, an outlier is an element of a data set which was generated according to a different process than the majority of the data. For instance, we may imagine that our samples $X_1, \ldots, X_n$ were sampled i.i.d. from a distribution $(1 - \varepsilon)D + \varepsilon N$ over $\mathbb{R}^d$, where $D$ is the distribution of inliers, $N$ is the distribution of outliers, and $\varepsilon > 0$ is a small number – that is, we imagine that a *constant fraction* of our data may be outliers.

For this discussion, we also informally imagine that $N$ is sufficiently distinct from $D$ that the set of outliers could be approximately identified by brute-force search over subsets of $(1 - \varepsilon)n$ samples, if given unlimited computational resources. Otherwise, outlier detection is not a meaningful problem, and robust mean estimation is easy (because the empirical mean will be a good estimator). Under these circumstances, what makes identifying outliers and estimating the mean in their presence difficult? Chiefly:

*Outliers may not be identifiable in isolation.* On its own, a typical outlier $X_i \sim N$ may look much like a typical inlier $X_j \sim D$. For instance, it could be $\|X_i\|_2 \approx \|X_j\|_2$, and $X_i, X_j$ may have similar distance to the nearest few neighboring samples, especially in high dimensions where samples are far apart.

*Outliers still introduce bias, collectively* Even if individual outliers look innocuous, the collective effect a modified $\varepsilon$-fraction of samples $X_i$ can still substantially change the empirical distribution of $X_1, \ldots, X_n$. As a result, even simple statistical tasks like estimating the mean or covariance of $D$ require sophisticated estimators: naively pruning individual outliers and then employing standard empirical estimators typically leads to far-suboptimal error rates. For example, an $\varepsilon$-fraction of $X_1, \ldots, X_n$ which are all slightly biased in a single direction may shift the empirical mean of $X_1, \ldots, X_n$, but this bias will be difficult to detect by looking at small numbers of samples at once. This also demonstrates that successful outlier detection can require global geometric information about a high-dimensional dataset, such as whether or not a direction exists in which many (say, $\varepsilon n$) samples are unusually biased.

*Outliers may be inhomogeneous.* Outliers need not exhibit unusual bias in only one direction, or all have the same norm, or lie in a single cluster. Rather, if a dataset exhibits several forms of corruption, there may be as many different-looking kinds of outliers. In the theoretical robust mean estimation setting, the adversary producing $\varepsilon$-corrupted samples may corrupt $\varepsilon n/10$ samples by biasing them in some direction, another $\varepsilon n/10$ samples by unusually enlarging their norms, and so forth.

Since robust mean estimation involves a malicious adversary, all of the above phenomena must be addressed by our robust mean estimation algorithm. In the empirical section of this paper, we focus on designing an outlier detection method suited to situations where at least one of them occurs – in other cases, existing methods (such as those based on Euclidean norms or local neighborhoods of individual samples [5]) may be more appropriate.

## 1.2 QUE: Quantum Entropy Scoring

Recent innovations in robust mean estimation [15, 8] rely on the following crucial observation about $\varepsilon$-corrupted samples $X_1, \ldots, X_n$ from a distribution $D$ with covariance $\Sigma \preceq \mathrm{Id}$. Namely: *any subset $S \subseteq \{X_1, \ldots, X_n\}$ of samples which shift the empirical mean by distance more than $\sqrt{\varepsilon}$ in some direction $v$ also introduce an eigenvalue of magnitude greater than $1$ to the empirical covariance.*

In robust mean estimation, this leads to (amongst others) the *filter* algorithm of [8, 9], one of the first to achieve dimension-independent error rates. Roughly speaking, the algorithm iterates the following until the empirical covariance $\overline{\Sigma}$ has small spectral norm: (1) compute the top eigenvector $v$ of the empirical covariance of $\overline{\Sigma}$, then (2) throw out samples $X_i$ whose projections $|\langle X_i - \overline{\mu}, v\rangle| \gg 1$ is unusually large, where $\overline{\mu}$ is the empirical mean of the corrupted dataset. For outlier detection this suggests a natural scoring rule – let the outlier score $\tau_i$ of sample $X_i$ be proportional to $|\langle X_i - \overline{\mu}, v\rangle|$.

The main drawback of these algorithms is that they do not adequately account for inhomogeneity of outliers. For the filter, this leads to a worst-case running time of $\widetilde{O}(nd^2)$, because the filter operation (which can be implemented in $\widetilde{O}(nd)$ time) may have to be repeated as many as $d$ times if the adversary introduces outliers lying in $d$ orthogonal directions. The rule $\tau_i = |\langle X_i - \overline{\mu}, v\rangle|$ may miss outliers causing a large eigenvalue of $\overline{\Sigma}$, but in a direction orthogonal to the top eigenvector $v$.

In the opposite extreme, if outliers are maximally inhomogeneous – no group of them is unusually biased in some shared direction $v$ – then the only way they can bias the empirical mean is for the individual $\ell_2$ norms $\|X_i - \overline{\mu}\|_2$ to be larger than typical. This suggests a different scoring rule: $\tau_i = \|X_i - \overline{\mu}\|_2$. This approach, however, breaks down in the situation we started with, that groups of outliers are biased in a shared direction but they do not have larger norms than good samples.

*Our main conceptual contribution is an approach to utilize information about outliers beyond what is available in the top eigenvector of the empirical covariance $\overline{\Sigma}$ and in individual $\ell_2$ norms.* Appropriately adapted to their respective settings, this leads to our algorithms for both robust mean estimation and outlier detection.

Our first observation is that *any eigenvalue/eigenvector $\lambda, v$ – not just the top ones – of the empirical covariance with $\lambda \gg 1$ must be due to outliers.* We therefore consider the intermediate goal of finding a *distribution* over directions $v \in \mathbb{R}^d$ containing information about as many outlier directions as possible. We formalize this as the following entropy-regularized convex program over $d \times d$ positive semidefinite matrices:

$$\max_{U \in \mathbb{R}^{d \times d}} \alpha \cdot \langle U, \overline{\Sigma}\rangle + S(U) \text{ such that } U \succeq 0, \mathrm{tr}(U) = 1 , \tag{1}$$

where $\alpha \geq 0$ is some constant and $\langle A, B\rangle = \mathrm{tr}(AB^\top)$ denotes the trace inner product of matrices. Here, $S(U) = -\langle U, \log U\rangle$ is the *quantum entropy* (also known as the *von Neumann entropy*) of the matrix $U$. If $U = \sum_{i=1}^d \mu_i v_i v_i^\top$ is the eigendecomposition of $U$, since it has $\mathrm{tr}(U) = 1$ we may interpret it as a distribution over orthonormal vectors $v_1, \ldots, v_d$ with weights $\mu_1, \ldots, \mu_d$ and hence with entropy $S(U)$. Under this interpetation, $\langle U, \overline{\Sigma}\rangle = \mathbb{E}_{v_i \sim \mu}\langle v_i, \overline{\Sigma} v_i\rangle$. As $\alpha$ varies, (1) trades off optimizing for a distribution supported on many distinct directions for a distribution supported on eigenvectors of $\overline{\Sigma}$ with large eigenvalues. The optimizer of (1) takes the form $U = \exp(\alpha \cdot \overline{\Sigma})/\mathrm{tr}\exp(\alpha \cdot \overline{\Sigma})$ where $\exp(\cdot)$ is the matrix exponential function.

**Definition 1.1.** Let $U = \exp(\alpha \cdot \overline{\Sigma})/\mathrm{tr}\exp(\alpha \cdot \overline{\Sigma})$ be the optimizer of (1), for some data set $\mathcal{X} = X_1, \ldots, X_n \in \mathbb{R}^d$ where $\overline{\Sigma}$ is the covariance of $\mathcal{X}$. The quantum entropy (QUE) scores with parameter $\alpha$ are given by $\tau_i = (X_i - \overline{\mu})^\top U(X_i - \overline{\mu})$, where $\overline{\mu}$ is the mean of $\mathcal{X}$.

Intuitively, the QUE scores will penalize any point which is causing a large eigenvalue in any direction, which should allow us to find more outliers than the naive spectral scores presented above. QUE scores also interpolate between two more naive scoring rules: when $\alpha = 0$ we have $U = \mathrm{Id}/d$ and so $\tau_i = \frac{1}{d}\|X_i - \overline{\mu}\|_2^2$ is the $\ell_2$ norm (up to a scaling), while when $\alpha \to \infty$ we have $U \to vv^\top$ where $v$ is the top eigenvector of $\overline{\Sigma}$, recovering naive spectral scoring. *In both experiments and theory we find that choosing $\alpha$ strictly between $0$ and $\infty$ outperforms either of the extreme choices.*

QUE scores are also appealing from a computational perspective: we show that a list of approximate QUE scores $\tau_i' = (1 \pm 0.01)\tau_i$ can be computed from $X_1, \ldots, X_n$ in nearly-linear time, by appropriate use of Johnson-Lindenstrauss sketching and efficient computation of the matrix exponential by

series expansion. This is crucial to both the nearly-linear running time of our algorithm for robust mean estimation and to the scalability of our outlier detection method.

In Section 1.4 we describe refinements of QUE scoring which fit it into the matrix multiplicative weights framework [3], leading to our nearly-linear time algorithms for robust mean estimation. We give two very similar algorithms, one for when the distribution of inliers is only assumed to have bounded covariance, and one when the inliers are assumed to be subgaussian. The resulting algorithms are conceptually similar to the following modification of the filter mentioned above: until $\|\overline{\Sigma}\|_2 \leq O(1)$, compute QUE scores $\tau_i$, throw out data points $X_i$ with $\tau_i \gg 1$, and repeat. (To obtain provable guarantees, our final algorithms are somewhat more complex: in some iterations we use QUE scores based on certain reweightings of the data learned in previous iterations.)

In Section 1.5 we describe experiments validating the QUE scoring rule on both synthetic and real data sets. We show that it performs especially well by comparison to local-neighborhood methods and to scoring based on only the top eigenvector in data sets where the inliers are close to isotropic (or can be made so by applying data whitening procedures) and in which there are heterogeneous outliers.

## 1.3 Related work

**Robust mean estimation:** The study of robust statistics and in particular robust mean estimation began with major works by Anscombe, Huber, Tukey and others in the 1960s [2, 25, 12, 26]. The literature on polynomial-time algorithms for robust statistics has exploded in recent years, following works by Diakonikolas et al and Lai, Rao and Vempala giving the first polynomial-time algorithms for robust mean estimation with dimension-independent (or nearly dimension-indepedent) error [8, 15]. A full survey is beyond our scope here – see e.g. the recent theses [17, 24] for a thorough account. Particularly relevant to our work is the recent work of Cheng, Diakonikolas, and Ge who design an algorithm for robust mean estimatin with running time $\widetilde{O}(nd/\varepsilon^6)$ – the first to achieve nearly linear time for constant $\varepsilon$ – by appeal to nearly linear time solvers for packing and covering semidefinite programs [6]. Our algorithms carry two advantages over this prior work: first, our algorithm runs in nearly linear time for any choice of $\varepsilon = \varepsilon(n, d)$, and second, because we avoid the $1/\varepsilon^6$ scaling and appeal to semidefinite programming, our theoretical ideas lead to a practical method for outlier detection. The techniques of Diakonikolas et al. were later extended to robust covariance estimation [7]; it remains an interesting direction to extend our techniques to covariance estimation.

*Concurrent work:* After this manuscript was initially submitted, we became aware of the concurrent work [16], which also obtains a nearly-linear time algorithm for robust mean estimation of distributions with bounded covariance. The algorithm of [16] also obtains *subgaussian confidence intervals* (see e.g. [19]), which the algorithm in this work does not. By contrast, the algorithms in our work also obtain improved rates of error with respect to $\varepsilon$ when the underlying distribution is sub-Gaussian, and our method is sufficiently practical that we are able to implement parts of it to run our experiments on outlier detection. (The method of [16] relies on nearly-linear time solvers for packing/covering semidefinite programs, which are not yet practical.) Finally, implicit in the work [16] is a reduction from arbitrary $\varepsilon$ to the case $\varepsilon = 1/100$; we describe this reduction and some consequences in supplementary material.

**Outlier detection** Detection of outliers goes back nearly to the beginning of statistics itself [11]. Even restricting to the high dimensional case it has a literature too broad to survey here. Much recent work has focused on so-called *local outlier factor*-based methods, which assign outlier scores based on the local density of other samples near each $X_i$ – see e.g. [13, 14] and further references in [5]. We find that QUE scoring compares favorably to such local methods in high-dimensional datasets like we describe in Section 1.1 – see Sections 1.5 and supplementary material for details.

## 1.4 Robust mean estimation: results and algorithm overview

We turn to our algorithm for robust mean estimation, deferring details to supplementary material.

**Definition 1.2** ($\varepsilon$-corrupted samples)**.** Let $D$ be a distribution on $\mathbb{R}^d$. We say that $X_1, \ldots, X_n$ are an $\varepsilon$-corrupted set of samples from $D$ if they are first drawn i.i.d. from $D$, then modified by an adversary who may adaptively inspect all the samples, remove $\varepsilon n$ of them, and replace them with arbitrary vectors in $\mathbb{R}^d$.

Note that $\varepsilon$-corruption is a *stronger* outlier model than the $(1-\varepsilon)D+\varepsilon N$ mixture model we described in Section 1; our algorithms also work in this milder mixture model. Our main theoretical result is:

**Theorem 1.1.** *For every $n, d \in \mathbb{N}$ and $\varepsilon > 0$ there are algorithms* QUESCOREFILTER *,S.G.- QUESCOREFILTER with running time $\widetilde{O}(nd)$, such that for every distribution $D$ on $\mathbb{R}^d$ with mean $\mu$ and covariance $\Sigma$, given $n$ $\varepsilon$-corrupted samples from $D$, QUESCOREFILTER produces $\hat{\mu}$ such that $\|\hat{\mu} - \mu\|_2 \leq O(\sqrt{\varepsilon}) + \widetilde{O}(\sqrt{d/n})$ if $\Sigma \preceq$ Id, and S.G.-QUESCOREFILTER produces $\hat{\mu}$ such that $\|\hat{\mu} - \mu\|_2 \leq O(\varepsilon\sqrt{\log(1/\varepsilon)} + \sqrt{d/n})$ if $D$ is sub-Gaussian with $\Sigma =$ Id, all with probability at least $0.99$.*

For the bounded covariance case, the $O(\sqrt{\varepsilon})$ term information-theoretically optimal up to constant factors. The other term, $\widetilde{O}(\sqrt{d/n})$, is information-theoretically optimal up to the logarithmic factors in the $\widetilde{O}(\cdot)$ even without corruptions. For the sub-Gaussian case, the $O(\varepsilon\sqrt{\log 1/\varepsilon})$ term is believed to be necessary for computationally efficient algorithms (see e.g the statistical-query lower bound [10]), although that term can be made $O(\varepsilon)$ by using computationally-intractable estimators such as Tukey median, and the latter is information-theoretically optimal [26]. The $\sqrt{d/n}$ term is information-theoretically optimal even without corruptions.

In this section we discuss our algorithm for the bounded-covariance case $\Sigma \preceq$ Id in the setting that the adversary may not remove samples, leaving technical details and the modifications necessary to handle removed samples and sub-Gaussian $D$ to supplementary material.

**Definition 1.3** (Simplified robust mean estimation). *Let $S = \{X_1, \ldots, X_n\} \subseteq \mathbb{R}^d$ be a dataset with the property that $S$ partitions into $S = S_g \cup S_b$ with $|S_b| \leq \varepsilon n$ and $\mathbb{E}_{i \sim S_g}(X_i - \mu_g)(X_i - \mu_g)^\top \preceq$ Id, where $\mu_g = \mathbb{E}_{i \sim S_g} X_i$. Given $S$, the goal is to find a vector $\hat{\mu}$ with $\|\mu_g - \hat{\mu}\|_2 \leq O(\sqrt{\varepsilon})$.*

Like prior algorithms for robust mean estimation, ours maintains a *weight vector* $w_1, \ldots, w_n \geq 0$ with $\sum w_i \leq 1$, initialized to $w_i = 1/n$. The algorithm iteratively decreases the weight of points suspected to be outliers that are causing $\|\mu(w) - \mu_g\|_2$ to be large.[2] A key insight of recent work on robust mean estimation is that it suffices to find weights $w$ which place almost as much mass on $S_g$ as does the uniform weighting and whose empirical covariance is small. This is formalized in the following lemma. For a weight vector $w$, let $|w| = \sum w_i$, $\mu(w) = \frac{1}{|w|}\sum w_i X_i$, and $M(w) = \frac{1}{|w|}\sum w_i(X_i - \mu(w))(X_i - \mu(w))^\top$. Let $\|M\|_2$ be the spectral norm of a matrix $M$.

**Lemma 1.2** (Implicit in prior work). *Let $S = \{X_1, \ldots, X_n\}$ be as in Definition 1.3. Suppose that $w$ is a weight vector such that $\|M(w)\|_2 \leq O(1)$ and $w$ is mostly good, by which we mean $|\frac{1}{n}\mathbf{1}_{S_g} - w_g| \leq |\frac{1}{n}\mathbf{1}_{S_b} - w_b|$, where $\mathbf{1}_{S_g}, \mathbf{1}_{S_b}$ are the indicators of $S_g, S_b$ and $w_g, w_b$ are $w$ restricted to $S_g, S_b$ respectively. (Intuitively, $w$ is mostly good if it results by removing from the uniform weighting $\mathbf{1}_S/n$ more weight from $S_b$ than from $S_g$.) Then $\|\mu(w) - \mu_g\|_2 \leq O(\sqrt{\varepsilon})$.*

Lemma 1.2 captures the following geometric intuition: if the bad points $S_b$ receive enough weight in $w$ to cause $\|\mu(w) - \mu_g\|_2 \gg \sqrt{\varepsilon}$, then an $O(\varepsilon)$-fraction of the mass of $w$ is on $X_i$ which are unusually correlated with the vector $\mu(w) - \mu_g$, which leads to a large maximum eigenvalue in $M(w)$. Prior works employ a variety of methods to find a mostly good weight vector $w$ with $\|M(w)\|_2 \leq O(1)$. Perhaps the simplest is the *filter* of [8], which iterates: While $\|M(w)\|_2 \gg 1$, compute its top eigenvector $v$ and *naive spectral scores* $\tau_i = \langle X_i - \mu(w), v\rangle^2$. Throw out $X_i$ with large $\tau_i$ and repeat.

The filter ensures that the weight vector it maintains is mostly good because (in an averaged sense) $\tau_i$ can be large only for $X_i$ which are corrupted. This is because the (weighted) sum of all scores $\sum w_i \tau_i = \langle M(w), vv^\top\rangle \gg 1$, while the contribution to this sum from $S_g$ has $\sum_{i \in S_g} w_i \tau_i \approx \langle \frac{1}{n}\sum_{i \in S_g}(X_i - \mu_g)(X_i - \mu_g)^\top, vv^\top\rangle \leq 1$. (Here we ignore some details about centering $X_i$ at $\mu_g$ rather than $\mu(w)$.) Thus, the $\tau_i$ from $S_b$ must make up almost all of $\sum w_i \tau_i$. Simple approaches to removing or downweighting $X_i$ with large $\tau_i$ then remove strictly more weight from $S_b$ than from $S_g$.

However, filtering based on naive spectral scores alone faces a barrier to achieving nearly-linear running-time. If the corruptions $S_b$ are split among many orthogonal directions, the naive spectral

filter will have to find those directions one at a time. Thus, it may require $\Omega(d)$ iterations (leading to $\Omega(nd^2)$ running time) to arrive at $w$ with $\|M(w)\|_2 \leq O(1)$.

Our main idea is that by replacing naive spectral scores with slightly modified QUE scores, each iteration of the filter can take into account projections of each sample onto many large eigenvectors of $M(w)$. We show that our modified QUE scores $\tau_i$ maintain the property that $\sum_{i \in S_b} w_i \tau_i \gg \sum_{i \in S_g} w_i \tau_i$, and so downweighting according to $\tau_i$ removes more mass from $S_b$ than $S_g$. However, filtering with QUE scores makes faster progress than with naive spectral scores: roughly speaking, we show that only $O(\log d)^2$ rounds of filtering according to QUE scores are required to find a mostly-good weight vector $w$ with $\|M(w)\|_2 \leq O(1)$.

The core of our algorithm is a subroutine, DECREASESPECTRALNORM, to take a mostly good weight vector $w$ with $\|M(w)\|_2 \gg 1$ and in $O(\log d)$ rounds of QUE filtering produce another mostly good $w'$ with $\|M(w')\|_2 \leq \frac{3}{4}\|M(w)\|_2$. Repeating this subroutine $O(\log d)$ times and then outputting the resulting $\mu(w)$ yields our main algorithm. An outline of this subroutine is presented as Algorithm 1. We first establish a rigorous sense in which downweighting according to outlier scores $\tau_i$ makes progress: *it decreases the weighted average of the scores while removing more weight from bad points than good*.

**Lemma 1.3** (Progress in one round of downweighting, informal)**.** *There is a downweighting algorithm which takes a density matrix $U$ and a mostly good weight vector $w$ and produces a mostly good weight vector $w'$ by downweighting points with large score $\tau_i = \langle X_i - \mu(w), U(X_i - \mu(w)) \rangle$ such that $\sum w_i' \tau_i \leq \frac{1}{3} \sum w_i \tau_i$ so long as $\sum w_i \tau_i \gg 1$. Furthermore, $M(w') \preceq M(w)$.*

Let us give a geometric interpretation to Lemma 1.3: it establishes that if $\sum w_i \tau_i = \langle U, M(w) \rangle \gg 1$ then the quadratic form of $M(w')$ decreases *in the directions defined by $U$*, since

$$\langle M(w'), U \rangle \approx \sum w_i' \tau_i \leq \frac{1}{3} \sum w_i \tau_i = \frac{1}{3} \langle M(w), U \rangle . \tag{2}$$

This guarantee becomes more meaningful as the entropy $S(U)$ increases, because it suggests the quadratic form of $M(w)$ has decreased in more directions. To make this formal, we appeal to the matrix multiplicative weights framework. DECREASESPECTRALNORM applies downweighting iteratively using a sequence of entropy-maximizing density matrices $U_1, \ldots, U_T$ chosen according to the matrix multiplicactive weights update rule, leading to a series of mostly good weight vectors $w_1, \ldots, w_T$ such that $\|M(w_T)\|_2 \leq \frac{3}{4}\|M(w_0)\|_2$. We choose

$$U_t = \exp \left( \frac{1}{\|M(w)\|_2} \sum_{k=0}^{t-1} M(w_k) \right) \Big/ \operatorname{tr} \exp \left( \frac{1}{\|M(w)\|_2} \sum_{k=0}^{t-1} M(w_k) \right) , \tag{3}$$

where $w_0 = w$ is the input weight vector, $U_0 = \mathrm{Id}$, and $w_t$ results from applying the downweighting of Lemma 1.3 to $w_{t-1}$ using $U_t$ (if $\langle M(w_{t-1}), U_t \rangle \gg 1$). The following lemma is a special case of the standard (local norm) *regret bound* for matrix multiplicative weights.

**Lemma 1.4** (Special case of Theorem 3.1, [1])**.** *For any $w_0, \ldots, w_T$, if $\alpha \leq 1/\|M(w_t)\|_2$ for all $t \leq T$, then*

$$\left\| \sum_{t=0}^{T-1} M(w_t) \right\|_2 \leq \sum_{t=0}^{T-1} \langle U_t, M(w_t) \rangle + \alpha \sum_{t=0}^{T-1} \langle U_t, M(w_t) \rangle \cdot \|M(w_t)\|_2 + \frac{\log d}{\alpha} . \tag{4}$$

Now we sketch the analysis of DECREASESPECTRALNORM.

**Claim 1.5** (Informal)**.** *If $w = w_0$ is mostly good, with $\|M(w_0)\|_2 \geq 100$, then DECREASESPECTRALNORM produces mostly good $w_T$ with $\|M(w_T)\|_2 \leq \frac{3}{4}\|M(w)\|_2$.*

*Proof sketch.* Since $M(w_t) \preceq M(w_{t+1})$ by Lemma 1.3, we have $\|M(w_t)\|_2 \leq \|M(w_0)\|_2$ for all $t$, and hence $\alpha = 1/\|M(w_0)\|_2 \leq 1/\|M(w_t)\|_2$ for all $t$, so $w_0, \ldots, w_T$ and $U_0, \ldots, U_{T-1}$ satisfy the hypotheses of Lemma 1.4. By our choice of $\alpha$ and $M(w_T) \preceq M(w_t)$ for all $t$, (4) implies

$$T \cdot \|M(w_T)\|_2 \leq \left\| \sum_{t=0}^{T-1} M(w_t) \right\|_2 \leq 2 \sum_{t=0}^{T-1} \langle U_t, M(w_t) \rangle + \|M(w_0)\|_2 \cdot \log d .$$

If $\langle U_t, M(w_{t-1})\rangle \geq \|M(w_0)\|_2/3 \gg 1$, then DECREASESPECTRALNORM performs down-weighting, and by Lemma 1.3 and (2) (which we establish rigorously in supplemental material), $\langle M(w_t), U_t\rangle \leq \frac{1}{3}\langle M(w_{t-1}), U_t\rangle \leq \frac{1}{3}\|M(w_0)\|$. Otherwise, by hypothesis $\langle M(w_t), U_t\rangle = \langle M(w_{t-1}), U_t\rangle \leq \|M(w_0)\|_2/3$. Using this bound and dividing by $T$, we obtain $\|M(w_T)\|_2 \leq (\frac{2}{3} + \frac{\log d}{T})\|M(w_0)\|_2$. Choosing $T \geq 20\log d$ completes the proof sketch. $\square$

*Running time:* Our overall algorithm only requires $\log(nd)^{O(1)}$ iterations of DECREASESPECTRAL-NORM, and the latter only requires $O(\log(d))$ iterations of downweighting, so we just have to implement downweighting in nearly-linear time. We show in supplemental material that this can be done by avoiding representing any of the matrices $U_t$ explicitly in memory: instead, we maintain only low-rank sketches of them. This leads to some approximation error in computing the QUE scores, but we show that approximations to the QUE scores suffice for all arguments above.

For remaining technical details and full proofs, see Sections 5-9 of supplemental materials.

---

**Algorithm 1** DECREASESPECTRALNORM

1: **Input:** $X_1, \ldots, X_n$ as in Definition 1.3, mostly good weight vector $w_0$.
2: For iteration $t = 0, \ldots, O(\log d)$, if $\|M(w_t)\|_2 \leq \frac{3}{4}\|M(w_0)\|_2$, output $w_t$ and halt. Otherwise, let $U_t$ as in (4). If $\langle U_t, M(w_{t-1})\rangle \leq \frac{1}{3}\|M(w_0)\|_2$, let $w_{t+1} = w_t$. Else let $w_{t+1}$ be the output of downweighting from Lemma 1.3 with $U_t$.
3: Output $w_T$.

---

### 1.5 Outlier detection: algorithm and experimental results

In this section, we empirically evaluate outlier detection using QUE scoring. QUE scoring can detect (some kinds of) *spectral outliers*. We call $X \in \mathbb{R}^d$ a spectral outlier with respect to a dataset $S$ if the list of squared projections $(\langle \overline{X}, v_1\rangle^2, \ldots, \langle \overline{X}, v_d\rangle^2)$ is atypical by comparison to most $Y \in S$, where $\overline{X} = X - \mathbb{E}_{Y \sim S} Y$ and $v_1, \ldots, v_d$ are the eigenvectors of the covariance matrix of $S$. The QUE scoring approach to aggregate the list $(\langle \overline{X}, v_1\rangle^2, \ldots, \langle \overline{X}, v_d\rangle^2)$ into one number carries (at least) two distinct advantages: first, the QUE scores of a dataset can be computed approximately in nearly-linear time, and second, the QUE scores weigh $\langle \overline{X}, v_i\rangle^2$ more heavily for larger $\lambda_i$, while still incorporating more information than $\langle \overline{X}, v_1\rangle^2$ (which is the naive spectral approach). There may be many other useful ways to go beyond the naive spectral approach to combine the projections $(\langle \overline{X}, v_1\rangle^2, \ldots, \langle \overline{X}, v_d\rangle^2)$ into a single outlier score – indeed, by varying $\alpha$ QUE scoring already provides a tuneable range of methods.

*Experimental setup:* We must work with data containing well-defined and known inliers and outliers so that we can compare our results to ground-truth. We generate such data sets in three distinct ways, leading to three main experiments. (In supplemental material we also study some outlier-detection data sets appearing in prior work [5].)

*Synthetic:* We create synthetic data sets in 128 dimensions and $10^3 - 10^4$ samples with an $\varepsilon$-fraction of inhomogeneous outliers in $k$ directions by sampling from a mixture of $k + 1$ Gaussians $(1 - \varepsilon)\mathcal{N}(0, \mathrm{Id}) + \sum_{i=1}^{k} \varepsilon_i[\frac{1}{2}\mathcal{N}(C\sqrt{k/\varepsilon} \cdot e_i, \sigma^2 \mathrm{Id}) + \frac{1}{2}\mathcal{N}(-C\sqrt{k/\varepsilon} \cdot e_i, \sigma^2 \mathrm{Id})]$, where $e_1, \ldots, e_k$ are standard basis vectors, with $C \approx 1$ and $\sigma \ll 1$. The outliers are the samples from $\mathcal{N}(\pm C\sqrt{k/\varepsilon}e_i, \sigma^2 \mathrm{Id})$. By varying $\varepsilon, k$ and the distribution $\varepsilon_1, \ldots, \varepsilon_k$ of outlier weights, we demostrate in this simplified model how max-entropy outlier scoring improves on baseline algorithms in the presence of inhomogeneous outliers. We choose the scaling $\sqrt{k/\varepsilon} \cdot e_i$ because then standard calculations predict that if $\varepsilon_i \approx \varepsilon/k$ the outliers from $\mathcal{N}(\pm C\sqrt{k/\varepsilon}e_i, \sigma^2 \mathrm{Id})$ will contribute an eigenvalue greater than 1 to the overall empirical covariance.

*Mixed – word embeddings:* We create a data set consisting of word embeddings drawn from several sources. Inliers are the 100-dimensional GloVe embeddings ([21]) of the words in a random $\approx 10^3$ word long section of a novel (we use *Sherlock Holmes*) and outliers are embeddings of the first paragraphs of $k$ featured Wikipedia articles from May 2019 [27].

*Perturbed – images:* We create a data set consisting of CIFAR10 images some of which have artificially-introduced *dead pixels*. Inliers are $\approx 4500$ random CIFAR images $X \in \{1, \ldots, 256\}^{1024}$

(restricted to the red color channel). Outliers are $\approx 500$ random CIFAR images, partitioned into groups $S_1, \ldots, S_k$, such that for each group $i$ a random coordinate $p_i \in \{1, \ldots, 1024\}$ and a random value $c_i \in \{1, \ldots, 256\}$ is chosen and for each $X \in S_i$ we set $X_{p_i} = c_i$.

**Metric:** All the methods we evaluate produce a vector of *scores* $\tau_1, \ldots, \tau_n \in \mathbb{R}$. We use the standard *ROCAUC* metric to compare these scores to a ground-truth partition $S = S_g \cup S_b$ into inlier and outlier sets. $\text{ROCAUC}(\tau_1, \ldots, \tau_n, S_b, S_g) = \Pr_{i \sim S_b, j \sim S_g}(\tau_i \geq \tau_j)$ is simply the probability that a randomly chosen outlier is scored higher than a random inlier.

**Baselines:** We compare QUE scoring to the following other scoring rules. $\ell_2$: $\tau_i = \|X_i - \overline{\mu}\|$ is the distance of $X_i$ to the empirical mean; *top eigenvector naive spectral:* $\tau_i = \langle X_i - \overline{\mu}, v \rangle^2$ where $v$ is the top eigenvector of the empirical covariance; *k-nearest neighbors (k-NN)* [22, 5] and *local outlier factor (LOF)* [4, 5] methods: $\tau_i$ is a function of the distances to its $k$ nearest neighbors; *isolation forest and elliptic envelope:* standard outlier detection methods as implemented in scikit-learn [23, 18, 20].

**Whitening:** Scoring methods based on the projection of data points $X_i$ onto large eigenvectors of the empirical covariance work best when those eigenvectors correspond to directions in which many outliers lie. In particular, if $\Sigma_g$, the covariance of $S_g$, itself has large eigenvalues then such spectral methods perform poorly. We assume access to a *whitening transformation* $W \in \mathbb{R}^{d \times d}$, which captures a small amount of prior knowledge about the distribution of inliers $S_g$. For best performance $W$ should approximate $W^* = (\Sigma_g)^{-1/2}$ since $W^* X_i$ form an isotropic set of vectors. Of course, to compute $W^*$ exactly would require knowing which points are inliers, but we find that relatively naive approximations suffice. In particular, if a clean dataset $Y_1, \ldots, Y_m$ whose distribution is similar to the distribution of inliers is available, its empirical covariance can be used to find a good whitening transformation $W$. In our synthetic data we use $W = \text{Id}$. In our word embeddings experiment, we obtain $W$ using the empirical covariance of the embedding of another random section of Sherlock Holmes. In our CIFAR-10 experiment, we obtain $W$ from the empirical covariance of a fresh sample of $\approx 5000$ randomly chosen images from CIFAR-10.

---

**Algorithm 2** QUE-Scoring for Outlier Detection

---

1: **Input:** dataset $X_1, \ldots, X_n \in \mathbb{R}^d$, optional whitening transformation $W \in \mathbb{R}^{d \times d}$, scalar $\alpha > 0$.
2: Let $X_i' = W X_i$ be whitened data, $\overline{\mu} = \frac{1}{n} \sum_{i=1}^n X_i'$ and $\overline{\Sigma} = \frac{1}{n} \sum_{i=1}^d (X_i' - \overline{\mu})(X_i' - \overline{\mu})^\top$.
3: For $i \leq n$, let $\tau_i = (X_i'^\top \exp(\alpha \overline{\Sigma}/\|\overline{\Sigma}\|_2) X_i')/ \text{Tr} \exp(\alpha \overline{\Sigma}/\|\overline{\Sigma}\|_2)$. Return $\tau_1, \ldots, \tau_n$.
*Note on $\alpha$:* in both synthetic and real data we find that $\alpha = 4$ is a good rule-of-thumb choice, consistently resulting in improved scores over baseline methods.

---

**High-dimensional scaling:** Implementing Algorithm 2 by explicitly forming the matrix $\overline{\Sigma}$ and performing a singular value decomposition (SVD) to compute $\exp(\alpha \overline{\Sigma})$ is feasible on relatively low-dimensional data ($d \approx 100$). See supplementary material for discussion and results of a nearly-linear time implementation.

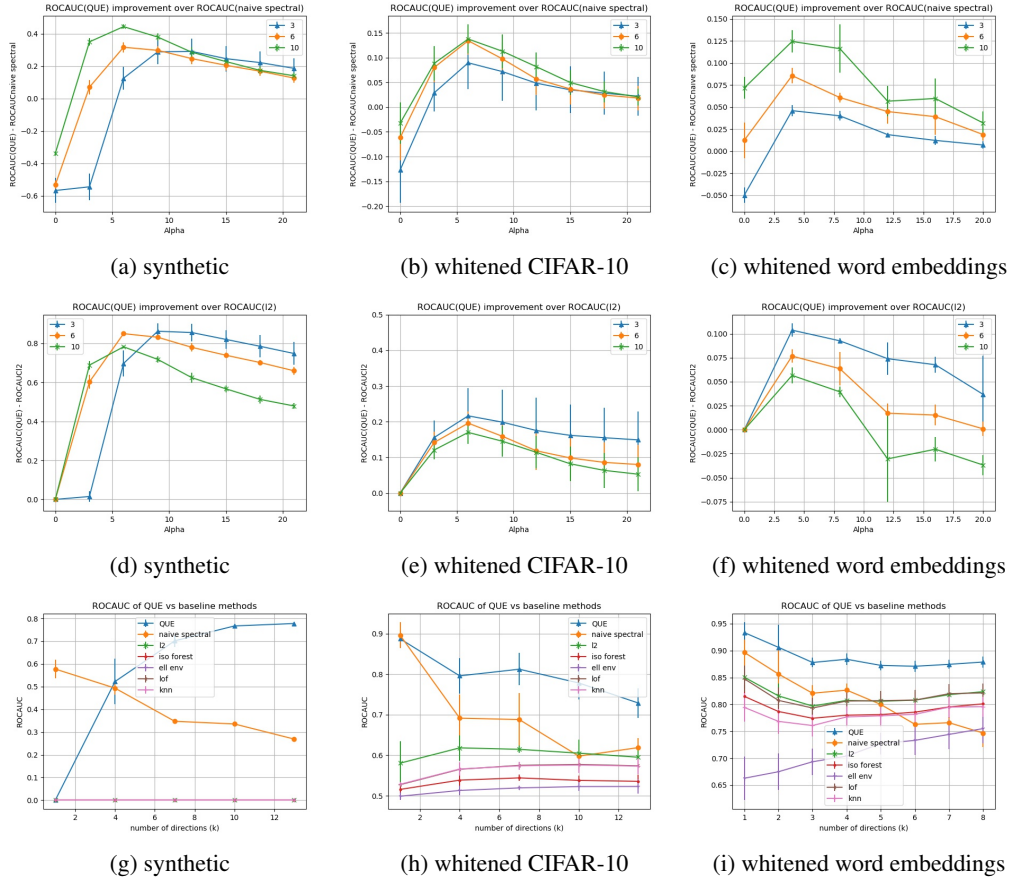

Figure 1: **(a-f)**: We plot the difference between ROCAUC performance of QUE and naive spectral (a-c), $\ell_2$ scoring (d-f) on all three data sets, as $\alpha$ varies. Error bars represent one empirical standard deviation in 20 trials. Note that in all three cases the mean improvement in ROCAUC score given by QUE is at least one standard deviation above 0 for a wide range of $\alpha$. Observe also that in synthetic data (which most closely parallels theory) the optimal $\alpha$ decreases with increasing number of outlier directions, in accord with the need to find a higher-entropy solution to (1). **(g-i)** We plot ROCAUC scores of QUE (with $\alpha = 4$) and a variety of other methods as the number of outlier directions increases. Error bars represent one standard deviation over $3 - 4$ trials. Number of trials is small due to large running time requirements of Scikit-learn methods IsolationForest and EllipticEnvelope. The methods "lof" and "knn" are based on nearest-neighbor distances [5]. All except spectral methods perform poorly on synthetic data; as $k$ increases the performance gap between QUE and naive spectral scoring grows. In all plots $\varepsilon = 0.2$. Experiments were generated on a quad-core 2.6Ghz machine with 16GB RAM and an NVIDIA P100 GPU.

## Footnotes

[1]Code is available at `https://github.com/twistedcubic/que-outlier-detection`.

[2] Some prior algorithms, e.g. the *filter* of [8] instead iteratively throw out points suspected to be outliers. However, since those algorithms are (necessarily) randomized, they can also be viewed as weighting points, where the weight of $X_i$ is the probability it has not been thrown out. The algorithm we present here can also be implemented by throwing out points in a randomized fashion – we discuss further in the appendix.

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
