[Supplementary Material]

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

 $(1 - \epsilon)\mathcal{N}(0, \mathrm{Id}) + \sum_{i=1}^{k} \epsilon_i(\frac{1}{2}\mathcal{N}(C\sqrt{k/\epsilon} \cdot e_i, \sigma^2 \mathrm{Id}) + \frac{1}{2}\mathcal{N}(-C\sqrt{k/\epsilon} \cdot e_i, \sigma^2 \mathrm{Id})$, where $e_1, \ldots, e_k$ are standard basis vectors, with $C \approx 1$ and $\sigma \ll 1$. The outliers are the samples from $\mathcal{N}(\pm C\sqrt{k/\epsilon}e_i, \sigma^2 \mathrm{Id})$. By varying $\epsilon, k$ and the distribution $\epsilon_1, \ldots, \epsilon_k$ of outlier weights, we demostrate in this simplified model how max-entropy outlier scoring improves on baseline algorithms in the presence of inhomogeneous outliers. We choose the scaling $\sqrt{k/\epsilon} \cdot e_i$ because then standard calculations predict that if $\epsilon_i \approx \epsilon/k$ the outliers from $\mathcal{N}(\pm C\sqrt{k/\epsilon}e_i, \sigma^2 \mathrm{Id})$ will contribute an eigenvalue greater than 1 to the overall empirical covariance.

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

# Roadmap of supplementary material

Here we provide a quick guide to the rest of the supplementary material. In Part I (Sections 4—9), we give omitted details to the description, and proof of correctness, of our nearly-linear time robust mean estimation algorithm. In Part II (Sections 10 and 11), we give additional experimental results and omitted details for the empirical evaluation of our outlier detection method based on QUE scoring.

# Part I: Nearly-linear time robust mean estimation

## 4 Preliminaries

For two functions $f, g$, we say $f = \widetilde{O}(g)$ if $f = O(g \log^c g)$ for some universal constant $c > 0$. We similarly define $\widetilde{\Omega}$ and $\widetilde{\Theta}$. For vectors $v \in \mathbb{R}^d$, we let $\|\cdot\|_2$ denote the usual $\ell_2$ norm, and $\langle \cdot, \cdot \rangle$ denote the usual inner product between vectors. Let $\mathbf{1}_m \in \mathbb{R}^m$ denote the $m$-dimensional all-ones vector.

For matrices $A, M \in \mathbb{R}^{d \times d}$ we let $\|M\|_2$ denote its spectral norm, we let $\|M\|_F$ denote its Frobenius norm, and we let $\langle M, A \rangle = \mathrm{tr}(M^\top A)$ denote the trace inner product between matrices. For any symmetric matrix $A \in \mathbb{R}^{d \times d}$, let $\exp(A)$ denote the usual matrix exponential of $A$. Finally, for scalars $x, y \in \mathbb{R}$, and any $\alpha > 0$, we say that $x \approx_\alpha y$ if $\frac{1}{1+\alpha} x \leq y \leq (1 + \alpha)x$.

We say a distribution $D$ over $\mathbb{R}^d$ is isotropic if $\mathrm{Cov}_{X \sim D}[X] = \mathrm{Id}$. We say a univariate distribution $D$ with mean $\mu$ is sub-gaussian with variance proxy $s^2$ if

$$\mathop{\mathbb{E}}_{X \sim D} \left[ (X - \mu)^k \right] \leq \mathop{\mathbb{E}}_{X \sim \mathcal{N}(0, s^2)} \left[ X^k \right]$$

for all $k$ even. We say a distribution $D$ over $\mathbb{R}^d$ and mean $\mu$ is sub-gaussian with variance proxy $\Sigma \preceq I$, if for all unit vectors $v$, the distribution of $\langle v, X - \mu \rangle$ is sub-Gaussian with variance proxy $v^\top \Sigma v$. Intuitively, a sub-Gaussian distribution is simply any distribution which concentrates as well as a Gaussian.

### 4.1 Our results

With this terminology in place, we are now ready to state our main results on robust mean estimation. Our first result is for robust mean estimation under the assumption of bounded covariance:

**Theorem 4.1.** *Let $D$ be a distribution with mean $\mu$ and covariance $\Sigma \preceq I$. Let $\varepsilon > 0$ be sufficiently small, and let $\delta > 0$. Let $S$ be an $\varepsilon$-corrupted set of samples from $D$ of size $n$. There is an algorithm* QUEScoreFilter$(S, \delta)$ *which takes $S$ and $\delta$, and outputs $\widehat{\mu}$ so that with probability $1 - \delta - \exp(-\varepsilon n)$, we have*

$$\|\mu - \widehat{\mu}\|_2 \leq O\left( \sqrt{\varepsilon} + \sqrt{\frac{d}{n\delta}} + \sqrt{\frac{d(\log d + \log 1/\delta)}{n}} \right) .$$

*Moreover, the algorithm runs in time $\widetilde{O}(nd \log 1/\delta)$.*

We make two observations about this problem. First, it is well-understood (see e.g. [12, 11]) that $\Omega(\sqrt{\varepsilon})$ is unavoidable for this problem, no matter how many samples are given. Second, observe that the rate $O(\sqrt{d/n})$ is necessary for this problem even without corruptions. Thus, up to log factors, and the dependence on $\delta$, this guarantee is information-theoretically optimal.

We also prove a strong statement for the case of robust mean estimation for sub-gaussian distributions:

**Theorem 4.2.** *Let $D$ be an isotropic sub-gaussian distribution with variance proxy $I$ and mean $\mu$. Let $\varepsilon > 0$ be sufficiently small, and let $\delta > 0$. Let $S$ be an $\varepsilon$-corrupted set of samples from $D$ of size $n$. There is*

an algorithm S.G.-QUESCOREFILTER$(S, \delta, \varepsilon)$ which takes $S, \delta$, and $\varepsilon$, and outputs $\widehat{\mu}$ so that with probability $1 - \delta$, we have

$$\|\mu - \widehat{\mu}\|_2 \leq O\left(\varepsilon\sqrt{\log 1/\varepsilon} + \sqrt{\frac{d + \log 1/\delta}{n}}\right) .$$

Moreover, the algorithm runs in time $\widetilde{O}(nd \log 1/\delta)$.

It is suspected, based on statistical-query lower bounds, that $\Omega(\varepsilon\sqrt{\log 1/\varepsilon})$ error is incurred by any computationally-efficient algorithm in this setting, although $\Theta(\epsilon)$ is the minimax optimal dependence of the error rate on $\epsilon$ [17, 10]. Moreover, $\Omega\left(\sqrt{(d + \log 1/\delta)/n}\right)$ is the minimax rate for mean estimation for Gaussians without noise. Thus, the error guarantees of this algorithm are minimax optimal up to constants and the factor $\sqrt{\log(1/\epsilon)}$.

This algorithm assumes that the distribution is isotropic. There is evidence that such an assumption is necessary to get error beyond $\sqrt{\varepsilon}$ using computationally efficient (i.e. poly-time) algorithms [26]. In our theorem statement above we also assume that the variance proxy is at most $I$. This is done for simplicity: it is easily verifiable that our algorithm works (with an appropriate scaling in front of the error guarantee) if the variance proxy is PSD upper bounded by $\sigma^2 I$ for any $\sigma^2$.

# 5 Additional technical preliminaries

Before we describe our techniques, we require a few additional algorithmic tools, which we describe here.

## 5.1 Soft selection of subsets of points

In our presentation of our filtering algorithm for robust mean estimation, it will be convenient for us to work with a "soft" version of the filter. Instead of wholly removing points that we deem suspicious, we will maintain a set of weights for each point, and downweight those that we find suspicious. In this section, we establish notation for dealing with such operations. However, we briefly remark that, as we will explain later in Appendix A.3, the same results (up to log factors) can be established using "hard" filtering more akin to the algorithms presented in prior work, e.g. [4].

Throughout this paper, we will let $\Delta_n$ denote the simplex in $n$ dimensions, and we let

$$\Gamma_n = \{w \in \mathbb{R}^n : w_i \geq 0, \sum w_i \leq 1\} .$$

For $w \in \Gamma_n$, let $|w| = \sum w_i$ be its $\ell_1$ norm. For $w, w' \in \Gamma_n$, we say $w' \leq w$ if $w_i' \leq w_i$ for all $i = 1, \dots, n$. Let $S = \{X_1, \dots, X_n\}$ be a (multi)-set of $n$ points. For any $w \in \Gamma_n$, we let $\mu(w) = \mu(S, w) = \frac{1}{|w|}\sum_{i=1}^n w_i X_i$, and we let

$$M(w) = M(S, w) = \sum_{i=1}^n w_i (X_i - \mu(w))(X_i - \mu(w))^\top .$$

Typically when the set $S$ is understood, we will omit the dependence on $S$ in the notation. For any set $T \subseteq S$, we let $\mu(T) = \mu(w)$ where $w \in \Gamma_n$ is the vector $w_i = 1/n$ for $i \in T$ and $w_i = 0$ otherwise, and similarly we let $M(T) = M(w)$.

## 5.2 Naive pruning

One primitive we will require will be the ability to removes points which are "obviously" outliers. It is well-known that there exist randomized nearly-linear time algorithms for achieving this. For completeness we prove this lemma in Appendix A.

**Lemma 5.1** (folklore). *Let $\varepsilon > 1/2$, and let $\delta > 0$. Let $S \subset \mathbb{R}^d$ be a set of $n$ points so that there exists a ball $B$ of radius $r$ and a subset $S' \subseteq S$ so that $|S'| \geq (1 - \varepsilon)n$, and $S' \subset B$. Then there is an algorithm* NAIVEPRUNE$(S, r, \delta)$ *which runs in time $O(nd \log 1/\delta)$ and with probability $1 - \delta$ outputs a set of points $T \subseteq S$ so that $S' \subseteq T$, and $T$ is contained in a ball of radius $4r$.*

In the case where the output of NAIVEPRUNE satisfies the conditions of the lemma, we say that NAIVEPRUNE *succeeds.*

## 5.3 The one-dimensional filter

An important algorithmic primitive for us will be an univariate soft outlier removal step. The sub-problem considered here is as follows: we are given a set of nonnegative scores $\tau_1, \ldots, \tau_m$, with the guarantee that there is a small subset $S \subseteq [m]$ so that $\sum_{i \in S} \tau_i > \frac{1}{2} \sum_{i=1}^{m} \tau_i$, that is, they contribute a majority of the mass of the points. The goal is to then either downweight (or remove) the overall set of scores scores in such a way so that more mass from $S$ is removed than from outside of $S$, or alternatively, more points are removed from $S$ than from outside $S$. An algorithm for achieving this via downweighting has already been described in [12], and a randomized algorithm that achieves the same sorts of guarantees with high probability by removing points is implicit in the filtering algorithm of [1, 4] (e.g. in Algorithm 3 in Appendix A of [4]). In this paper, we will require a slight strengthening of these algorithms. We require that not only do we remove more weight from the bad points than the good points, but we also decrease the overall sum by a constant factor. We observe that while we will present a method for acheiving this via downweighting, one can achieve the same guarantee (with high probability) by removing points. In the main text, we choose to present the soft downweighting method for robust mean estimation for simplicity. See Appendix A.3 for details.

Formally, we describe an algorithm 1DFILTER and prove the following guarantee for the algorithm. The algorithm and its analysis are fairly straightforward so we defer the formal descriptions and proofs to Appendix A.2.

**Theorem 5.2.** *Let $\eta \in (0, 1/2)$, let $b \geq 2\eta$, and let $w_1, \ldots, w_m$ and $\tau_1, \ldots, \tau_m$ be non-negative numbers so that $\sum_{i=1}^{m} w_i \leq 1$. Let $\tau_{\max} = \max_{i \in [m]} \tau_i$. Suppose there exist two disjoint sets $S_g, S_b$ so that $S_g \cup S_b = [m]$, and moreover,*

$$\sum_{i \in S_g} w_i \tau_i \leq \eta \sigma , \text{ where } \sigma = \sum_{i=1}^{n} w_i \tau_i .$$

*Then* 1DFILTER$(w, \tau, b)$ *runs in time $O\left(\left(1 + \log \frac{\tau_{\max}}{b\sigma}\right) m\right)$ and outputs $0 \leq w' \leq w$ so that:*

- *more weight is removed from $S_b$ than $S_g$, i.e. $\sum_{i \in S_g} w_i - w'_i \leq \sum_{i \in S_b} w_i - w'_i$, and*

- *the weighted sum of the $\tau$ has decreased, i.e. $w'$ satisfies*

$$\sum_{i=1}^{m} w'_i \tau_i \leq b\sigma . \tag{5}$$

In particular, note that if $b = \Omega(1)$ and $\tau_{\max}/\sigma \leq (nd)^{O(1)}$, then this algorithm runs in nearly linear time.

**Randomized outlier removal** As mentioned previously, there is also a randomized strategy that avoids downweighting and achieves the same guarantee with high probability (up to logarithmic factors in runtime). Our overall robust mean estimation algorithm (for both settings presented in the paper) can be instantiated using this algorithm rather than 1DFILTER. While as far as we know this yields no theoretical improvements for robust mean estimation (indeed, our analysis of it proves bound which are worse by logarithmic factors than our analysis of soft downweighting), it is much closer to the practical outlier detection method used in the experiments in Section 3 and also to prior algorithms presented in [4], and may be of instructive value. For this reason we describe this algorithm in Appendix A.3.

## 5.4 Matrix Multiplicative weights

We will use the following form of the MMW update, which is essentially the same as presented in [18]. In each iteration $t = 0, \ldots, T$, the player chooses an action $U_k \in \Delta_{d \times d}$, receives a gain matrix $F_t \in \mathbb{R}^{d \times d}$, and receives reward $\langle F_k, U_k \rangle$. Then the player sees $F_k$. In [18], they demonstrate that if the player plays according to the entropy regularizer (or equivalently, matrix multiplicative weights), namely,

$$U_k = \exp\left( cI + \alpha \sum_{t=0}^{k-1} F_t \right) \ , \tag{6}$$

where $c$ is a constant ensuring that $\operatorname{tr}(X_k) = 1$, and $\alpha$ satisfies $\alpha F_t \preceq I$ for all $0 = 1, \ldots, T-1$, then we have that for any $U \in \Delta_{n \times n}$,

$$\sum_{t=0}^{T-1} \langle F_t, U - U_t \rangle \leq \alpha \sum_{t=0}^{T-1} \langle U_t, |F_t| \rangle \cdot \|F_t\|_2 + \frac{\log n}{\alpha} \ . \tag{7}$$

Here, for any symmetric matrix $A = \sum_{i=1}^d \lambda_i v_i v_i^\top$, we let $|A|$ denote $|A| = \sum_{i=1}^d |\lambda_i| v_i v_i^\top$. Equivalently, by rearranging terms, and taking a supremum over $U$ of (8), we obtain that the update satisfies

$$\left\| \sum_{t=0}^{T-1} F_t \right\|_2 \leq \sum_{t=0}^{T-1} \langle U_t, F_t \rangle + \alpha \sum_{t=0}^{T-1} \langle U_t, |F_t| \rangle \cdot \|F_t\|_2 + \frac{\log n}{\alpha} \ . \tag{8}$$

# 6 An MMW algorithm for robust mean estimation with bounded covariance

In this section we describe the algorithm which achieves Theorem 4.1. We first identify a deterministic condition on the set of inliers under which our algorithm is guaranteed to be correct. It is a very mild condition: at a high level, it simply states that the empirical mean of the samples is converging to the true mean, and the empirical covariance is bounded.

**Definition 6.1.** We say a set of points $S$ is $(\gamma_1, \gamma_2)$-good with respect to a distribution $D$ with mean $\mu$ and covariance $\Sigma \preceq \operatorname{Id}$ if the following two properties hold:

- $\|\mu(S) - \mu\|_2 \leq \gamma_1$, and

- $\|\operatorname{Cov}(S)\|_2 \leq \gamma_2$.

The following is a generalization of Lemma A.18 in [4], which states that, with high probability, any set of i.i.d. points from a distribution with bounded covariance will contain a large set which is good with respect to that distribution. For completeness we prove this lemma in Appendix B.

**Lemma 6.1.** *Let $\varepsilon \in [0, 1/2)$, and let $n$ be a positive integer. Let $D$ be a distribution with mean $\mu$ and covariance $\Sigma \preceq \operatorname{Id}$, and let $X_1, \ldots, X_n$ be independent draws from $D$. Then, with probability $1 - \delta - \exp(-\varepsilon n)$, there exists a set $S \subseteq \{X_1, \ldots, X_n\}$ so that the following two conditions are simultaneously satisfied:*

- *$|S| \geq (1 - \varepsilon)n$, and*

- *$S$ is $(\gamma_1, \gamma_2)$-good with respect to $D$, where*

$$\gamma_1 = \frac{1}{1 - \varepsilon} \cdot \left( \sqrt{\frac{2d}{n\delta}} + \sqrt{c\varepsilon} \right) \ , \text{ and } \gamma_2 = \frac{1}{1 - \varepsilon} \cdot \frac{d(\log d + \log 2/\delta)}{c'\varepsilon n} \ , \tag{9}$$

*for some universal constants $c, c' > 0$.*

In particular, observe that for constant $\delta$ and for $n = \Omega(d \log d / \varepsilon)$, we have that $\gamma_1 = O(\varepsilon)$ and $\gamma_2 = O(1)$.

## 6.1 The good set

Throughout we will let $S = S_g \cup S_b \setminus S_r$, where $S_g$ is $(\gamma_1, \gamma_2)$-good with respect to $D$, and $|S_b|, |S_r| \le \varepsilon|S|$. For any $w \in \Gamma_n$, we let $w_g \in \Gamma_{|S_g|}$ denote the restriction of $w$ to the indices in $S_g$, and similarly define $w_b \in \Gamma_{|S_b|}$.

Our set of weights of interest will be slightly different than those considered in prior papers, but morally captures the same concept, up to issues of reweighting. We will always guarantee that the weights we consider lie within the following set:

$$\mathfrak{S}_{n,\varepsilon} = \left\{ w \in \Gamma_n : w \le \frac{1}{n}\mathbf{1}_n \text{ and } \left| \frac{1}{n}\mathbf{1}_{|S_g|} - w_g \right| \le \left| \frac{1}{n}\mathbf{1}_{|S_b|} - w_b \right| \right\}.$$

Intuitively, weights in the set $\mathfrak{S}_{n,\varepsilon}$ are what happens when we remove points from our data set, but ensure that we always remove at least as much mass from the bad set as we do from the good set.

## 6.2 Geometric lemmata

We first prove the following sequence of structural lemmata. The first, which is implicit in the earlier work, and which is in some sense the fundamental geometric fact which guides our algorithmic design, gives an upper bound on the deviation between the weighted empirical mean of the data set and the true mean of the distribution in terms of the spectral norm of the weighted covariance of the dataset.

**Lemma 6.2.** *Let $S, \gamma_1, \gamma_2$ be as above, and let $w \in \mathfrak{S}_{n,\varepsilon}$. Then*

$$\|\mu(w) - \mu\|_2 \le \frac{1}{1 - 2\varepsilon} \cdot \left( \sqrt{\varepsilon\gamma_2} + (1 + \varepsilon)\gamma_1 + 2\sqrt{\varepsilon\|\Sigma(w)\|_2} \right).$$

*Proof.* Let $\rho = \mu(w) - \mu$. We have the following sequence of identities:

$$\|\mu(w) - \mu\|_2^2 = \langle \mu(w) - \mu, \mu(w) - \mu \rangle$$

$$= \frac{1}{|w|} \sum_{i=1}^{n} w_i \langle X_i - \mu, \rho \rangle$$

$$= \frac{1}{|w|} \left( \sum_{i \in S_g} w_i \langle X_i - \mu, \rho \rangle + \sum_{i \in S_b} w_i \langle X_i - \mu, \rho \rangle \right)$$

$$= \frac{1}{|w|} \left( \underbrace{\sum_{i \in S_g} \frac{1}{n} \langle X_i - \mu, \rho \rangle}_{W_1} - \underbrace{\sum_{i \in S_g} \left( \frac{1}{n} - w_i \right) \langle X_i - \mu, \rho \rangle}_{W_2} + \underbrace{\sum_{i \in S_b} w_i \langle X_i - \mu, \rho \rangle}_{W_3} \right).$$

We now upper bound each term separately. By Cauchy-Schwarz, we have

$$|W_1| \le (1 - \varepsilon) \left\| \frac{1}{(1 - \varepsilon)n} \sum_{i \in S_g} X_i - \mu \right\|_2 \|\rho\|_2$$

$$\le \gamma_1 \|\rho\|_2,$$

by the goodness of $S_g$.

We now turn our attention to $W_2$ and $W_3$. Both bounds will follow from the following claim:

**Claim 6.3.** *Let $w', \alpha \in \Gamma_n$ be so that $|w'| \le \varepsilon$, $w'_i \le \frac{1}{n}$ for all $i \in [n]$, and $w' \le \alpha$. Then, for any $v \in \mathbb{R}^d$, we have*

$$\left| \sum_{i=1}^{n} w'_i \langle X_i - \mu, v \rangle \right| \le \sqrt{\varepsilon\|\Sigma(\alpha)\|_2}\|v\|_2 + \varepsilon\|\mu(\alpha) - \mu\|_2\|v\|_2.$$

*Proof.* We have

$$\left| \sum_{i=1}^{n} w_i' \left\langle X_i - \mu, v \right\rangle \right| = \left| \sum_{i=1}^{n} w_i' \left\langle X_i - \mu(\alpha), v \right\rangle + |w'| \langle \mu(\alpha) - \mu, v \rangle \right|$$

$$\leq \left| \sum_{i=1}^{n} w_i' \left\langle X_i - \mu(\alpha), v \right\rangle \right| + \varepsilon \left| \langle \mu(\alpha) - \mu, v \rangle \right|$$

$$\leq \left| \sum_{i=1}^{n} w_i' \left\langle X_i - \mu(\alpha), v \right\rangle \right| + \varepsilon \left\| \mu(\alpha) - \mu \right\|_2 \left\| v \right\|_2 .$$

By Hölder's inequality, we have

$$\left( \sum_{i=1}^{n} w_i' \left\langle X_i - \mu(\alpha), v \right\rangle \right)^2 \leq \left( \sum_{i=1}^{n} \frac{(w_i')^2}{\alpha_i} \right) \cdot \left( \sum_{i=1}^{n} \alpha_i \left\langle X_i - \mu(\alpha), v \right\rangle^2 \right)$$

$$\overset{(a)}{\leq} |w'| \cdot \sum_{i=1}^{n} \alpha_i \left\langle X_i - \mu(\alpha), v \right\rangle^2$$

$$\overset{(b)}{\leq} \varepsilon \| \Sigma(\alpha) \|_2 \| v \|_2^2 .$$

Here (a) follows since $w_i' \leq \alpha_i$, and (b) follows from the definition of spectral norm, and the assumption on $|w'|$. Thus, by taking square roots and combining terms, we have

$$\left| \sum_{i=1}^{n} w_i' \left\langle X_i - \mu, v \right\rangle \right| \leq \sqrt{\varepsilon \| \Sigma(\alpha) \|_2} \| v \|_2 + \varepsilon \left\| \mu(\alpha) - \mu \right\|_2 \left\| v \right\|_2 ,$$

as claimed. $\qquad\square$

With this claim, we can now bound $W_2$ and $W_3$. To bound $W_2$, let $w_i' = \frac{1}{n} - w_i$ for $i \in S_g$ and $w_i = 0$ otherwise, and let $\alpha_i = \frac{1}{n}$ if $i \in S_g$, and 0 otherwise. Then, applying the claim with $v = \rho$ yields that

$$|W_2| \leq \sqrt{\varepsilon \| \Sigma(\alpha) \|_2} \| \rho \|_2 + \varepsilon \left\| \mu(S_g) - \mu \right\|_2 \| \rho \|_2 \leq \sqrt{\varepsilon \gamma_2} \cdot \| \rho \|_2 + \varepsilon \gamma_1 \| \rho \|_2 .$$

Similarly, to bound $W_3$, let $w_i' = w_i$ if $i \in S_b$ and $w_i' = 0$ otherwise, and let $\alpha_i = w_i$ for all $i \in S$. Again, letting $v = \rho$, we get that

$$|W_3| \leq \sqrt{\varepsilon \| \Sigma(w) \|_2} \| \rho \|_2 + \varepsilon \| \rho \|_2^2 ,$$

as well. Combining these three bounds, and using the fact that $|w| \leq 1$, yields that

$$\| \rho \|_2^2 \leq \sqrt{\varepsilon \gamma_2} \| \rho \|_2 + (1 + \varepsilon) \gamma_1 \| \rho \|_2 + 2 \sqrt{\varepsilon \| \Sigma(w) \|_2} \| \rho \|_2 + 2 \varepsilon \| \rho \|_2^2 .$$

Simplifying this expression then yields the desired bound on $\| \rho \|_2$. $\qquad\square$

We also require the following linear algebraic fact:

**Lemma 6.4.** *Let $w', w \in \Gamma_n$ so that $w' \leq w$. Then $\Sigma(w') \preceq \Sigma(w)$.*

*Proof.* We first observe that if $w' \leq w$, then

$$\sum_{i=1}^{n} w_i' (X_i - \mu(w))(X_i - \mu(w))^\top \preceq \sum_{i=1}^{n} w_i (X_i - \mu(w))(X_i - \mu(w))^\top = \Sigma(w) .$$

To complete the argument, observe that

$$\sum_{i=1}^{n} w_i'(X_i - \mu(w))(X_i - \mu(w))^\top = \sum_{i=1}^{n} w_i'(X_i - \mu(w'))(X_i - \mu(w'))^\top + |w'|(\mu(w') - \mu(w))(\mu(w') - \mu(w))^\top$$

$$\succeq \sum_{i=1}^{n} w_i'(X_i - \mu(w'))(X_i - \mu(w'))^\top = \Sigma(w') \ ,$$

and so $\Sigma(w') \preceq \Sigma(w)$, as claimed. $\qquad\square$

## 6.3   General algorithm description, bounded second moment

In this section we describe the algorithm we would like to run via matrix multiplicative weights, and demonstrate that it will terminate in a small number of iterations, when given good approximations to the entropic scores.

The algorithm, which we call QUESCOREFILTER , proceeds in epochs, and takes as input a corrupted dataset $S$, and a *score oracle* $\mathscr{O}$. Initially, in epoch $s = 0$, we let $w^{(0)} = \frac{1}{n}\mathbf{1}_n$. Then, in epoch $s$, the algorithm proceeds iteratively as follows. First, approximately compute $\lambda^{(s)} \approx_{0.1} \|M(w^{(s)})\|_2$, and if $\lambda^{(s)} \leq 100\gamma_2$, then we terminate and output $\mu(w^{(s)})$.

Otherwise, we let $w_0^{(s)} = w^{(s)}$. Then, in iteration $t = 0, \ldots, T_s$, we first approximately compute $\lambda_t^{(s)} \approx_{0.1}$ $\|M(w_t^{(s)})\|_2$. If $\lambda_t^{(s)} \leq \frac{2}{3}\lambda_0^{(s)}$, we terminate the epoch and let $w^{(s+1)} = w_t^{(s)}$. Otherwise, we let $U_t^{(s)}$ be prescribed by the MMW update with parameter $\alpha^{(s)} = \frac{1}{1.1 \cdot \lambda_0^{(s)}}$, where the loss is given as follows. At time $t$, and for all $i \in [n]$, we let $\tilde{\tau}_{t,i}^{(s)} = \mathscr{O}(S, w_0^{(s)}, \ldots, w_n^{(s)})$ be the set of scores that the oracle produces. We then compute $\sum_i \tilde{\tau}_{t,i}^{(s)}$. If $\sum_i \tilde{\tau}_{t,i}^{(s)} \leq \frac{1}{5}\lambda_0^{(s)}$, then let $w_{t+1}^{(s)} = w_t^{(s)}$. Otherwise, let $w_{t+1}^{(s)} = \text{1DFILTER}(w_t^{(s)}, \tilde{\tau}_t^{(s)}, 1/4)$. In either case, the algorithm receives the gain matrix $F_t = M(w_{t+1}^{(s)})$. The formal pseudocode for this algorithm is given in Algorithm 3.

**The score oracles**   There are two important score oracles for our purposes. The first is the *exact* score oracle $\mathscr{O}_{\text{exact}}$, whose output is

$$\tau_{t,i}^{(s)} = \left(X_i - \mu\left(w_{t,i}^{(s)}\right)\right)^\top U_t^{(s)} \left(X_i - \mu\left(w_{t,i}^{(s)}\right)\right) \ , \tag{10}$$

where $U_t^{(s)}$ is as in Algorithm 3, namely,

$$U_t^{(s)} = \exp\left(c \operatorname{Id} - \alpha \sum_{i=0}^{t-1} M(w_i^{(s)})\right) \ , \tag{11}$$

where $c$ is chosen so that $\operatorname{tr}(U_t^{(s)}) = 1$.

Filtering using these scores corresponds to the intuition that we are using $U_t^{(s)}$ as a high entropy certificate. Ignoring runtime, this set of scores would be the most natural set of scores for the algorithm. However, computing this score is quite inefficient (polynomial but super-linear). Thus, we also require an *approximate* score oracle, which is an oracle $\mathscr{O}_{\text{approx}}$ whose output $\tilde{\tau}_{t,i}^{(s)}$ satisfies that

$$\tilde{\tau}_{t,i}^{(s)} \approx_{0.1} \tau_{t,i}^{(s)} \tag{12}$$

for all $t, i, s$, where $\tau_{t,i}^{(s)}$ is defined in (10) (the choice of 0.1 in the approximation here as well as for $\lambda_t^{(s)}$ is arbitrary; any constant sufficiently small will suffice). In Section 8 we will demonstrate how to construct such an approximate score oracle in nearly-linear time.

---

**Algorithm 3** MMW-based filtering method for robust mean estimation with bounded second moments

---

1: **Input:** dataset $S \subset \mathbb{R}^d$ of size $n$, parameters $\gamma_1, \gamma_2$, score oracle $\mathscr{O}$
2: Let $w^{(0)} = \frac{1}{n}\mathbf{1}_n$.
3: **for** epoch $s = 0, \ldots, O(\log \kappa)$ **do**
4:      Let $\lambda^{(s)} \approx_{0.1} \left\| M(w^{(s)}) \right\|_2$
5:      **if** $\lambda^{(s)} \leq 100\gamma_2$ **then**
6:         **return** $\mu(w^{(s)})$
7:      **end if**
8:      Let $w_0^{(s)} = w^{(s)}$
9:      Let $\alpha^{(s)} = \frac{1}{1.1 \cdot \lambda_0^{(s)}}$
10:     **for** iteration $t = 0, \ldots, O(\log d)$ **do**
11:       Compute $\lambda_t^{(s)} \approx_{0.1} \|M(w_t^{(s)})\|_2$.
12:       **if** $\lambda_t^{(s)} \leq \frac{2}{3}\lambda_0^{(s)}$ **then**
13:         terminate epoch
14:       **end if**
15:       Let $U_t^{(s)}$ be given by MMW update with parameter $\alpha^{(s)}$
16:       For $i = 1, \ldots, n$, let $\tilde{\tau}_{t,i}^{(s)} = \mathscr{O}(S, w_0^{(s)}, \ldots, w_n^{(s)})$
17:       **if** $\sum_i \tilde{\tau}_{t,i}^{(s)} \leq \frac{1}{5}\lambda_0^{(s)}$ **then**
18:         Let $w_{t+1}^{(s)} = w_t^{(s)}$.
19:       **else**
20:         Let $w_{t+1}^{(s)} = 1\text{DF{\sc ilter}}(w_t^{(s)}, \tilde{\tau}_t^{(s)}, 1/4)$.
21:       **end if**
22:       Let feedback matrix to MMW update be $F_t = M(w_{t+1}^{(s)})$
23:     **end for**
24:     Let $w^{(s+1)} = w_t^{(s)}$
25: **end for**

---

## 6.4 Correctness of QUEScoreFilter

The remainder of this section is dedicated to a proof of correctness of QUESCOREFILTER when instantiated with $\mathscr{O}_{\text{approx}}$. Formally, we show:

**Theorem 6.5.** *Let $D$ be a distribution with covariance $\Sigma \preceq I$ and mean $\mu$. Let $\varepsilon < c$, where $c$ is a universal constant, and let $\gamma_1, \gamma_2 > 0$. Let $S$ be a dataset so that $S = S_g \cup S_b \setminus S_r$ so that $S_g$ is $(\gamma_1, \gamma_2)$-good with respect to $D$, and $|S_b|, |S_r| \leq \varepsilon|S|$. Suppose moreover that $\|X_i\|_2 \leq \kappa$ for all $i \in S$. Then QUESCOREFILTER$(S, \mathscr{O}_{\text{approx}})$ terminates after at most $S = O(\log \kappa)$ epochs, and outputs a $w \in \mathfrak{S}_{n,\varepsilon}$ so that*

$$\|\mu(w) - \mu\|_2 \leq O(\sqrt{\varepsilon\gamma_2} + \gamma_1).$$

*Moreover, each epoch runs for at most $O(\log d)$ iterations, requires $O(\log d)$ calls to $\mathscr{O}_{\text{approx}}$, and requires $\widetilde{O}(nd + n\log \kappa)$ additional computation.*

We first prove correctness. The main lemma is the following per-epoch guarantee:

**Lemma 6.6.** *The following invariants always hold. For all epochs $s$, we have:*

- *$w^s \in \mathfrak{S}_{n,\varepsilon}$, and*

- *If $\|M(w^{(s)})\|_2 > \frac{100}{1.1}\gamma_2$, then epoch $s$ finishes after $O(\log d)$ iterations, and outputs $w^{(s+1)}$ so that $\|M(w^{(s+1)})\|_2 \leq \frac{2}{3}\|M(w^{(s)})\|_2$.*

We first show how the lemma implies the theorem.

*Proof of Theorem 6.5 given Lemma 6.6.* We first prove correctness. The bound on the $\ell_2$ norm of $X_i$ for all $i \in S$ immediately also implies that $\|M(w)\|_2 \leq O(\kappa^2)$ for all $w \in \Gamma_n$, and in particular for $w = \frac{1}{n}\mathbf{1}_n$. By Lemma 6.6, after $S = O(\log \kappa)$ epochs the algorithm must terminate, and moreover every epoch can run for at most $O(\log d)$ iterations. Lemma 6.6 additionally implies that if $w$ is the output of QUESCOREFILTER$(S)$, then $w \in \mathfrak{S}_{n,\varepsilon}$, and moreover, $\|\Sigma(w)\|_2 \leq 110\gamma_2$. Therefore, by Lemma 6.2, we have $\|\mu(w) - \mu\|_2 \leq O(\sqrt{\varepsilon\gamma_2} + \gamma_1)$, from which the desired conclusion immediately follows.

We now prove the runtime bound. In every iteration, besides the call the the oracle, the only costly operations are the approximate top eigenvalue computations and running 1DFILTER. However, the approximate top eigenvalue computations can be done in time $\widetilde{O}(nd)$ via power method since we only ask for a constant multiplicative approximation, and the bound on $\|X_i\|_2$ implies that 1DFILTER runs in $O(n \log \kappa)$ time. $\quad\square$

We now prove Lemma 6.6.

*Proof.* Clearly these invariants hold at the beginning of the algorithm. Since in the remainder of the proof we will only deal with a single epoch $s$, for conciseness we will omit the superscript. We will require the following claim:

**Claim 6.7.** *Suppose $w \in \mathfrak{S}_{n,\varepsilon}$ so that $\|M(w)\|_2 \geq 100\gamma_2$, and let $U \in \Delta_{d \times d}$. Let $\tau_i = (X_i - \mu(w))^\top U(X_i - \mu(w))$, and $\tilde{\tau}_i$ be so that $\tilde{\tau}_i \approx_{0.1} \tau_i$ for all $i$. Suppose that $\lambda \approx_{0.1} \|M(w)\|_2$, and $\sum_{i=1}^n \tilde{\tau}_i \geq \frac{1}{5}\lambda$. Then, if $w' = 1\text{DFILTER}(w, \tilde{\tau}, 1/4)$, then $w \in \mathfrak{S}_{n,\varepsilon}$, and $\langle M(w'), U \rangle \leq 0.31 \langle M(w), U \rangle$.*

*Proof.* We first show that $\sum_{i \in S_g} w_i \tau_i \leq c \sum_{i=1}^n w_i \tau_i$ for some universal constant $c \leq 0.11$. Let $\widetilde{w}_i = \frac{1}{n}$ if $i \in S_g$ and $\widetilde{w}_i = 0$ otherwise. Then, we have

$$
\begin{aligned}
\sum_{i \in S_g} w_i \tau_i &= \left\langle \sum_{i \in S_g} w_i (X_i - \mu(w))(X_i - \mu(w))^\top, U \right\rangle \\
&\overset{(a)}{\leq} \left\langle \sum_{i=1}^n \widetilde{w}_i (X_i - \mu(w))(X_i - \mu(w))^\top, U \right\rangle \\
&= \left\langle \sum_{i=1}^n \widetilde{w}_i (X_i - \mu(\widetilde{w}))(X_i - \mu(\widetilde{w}))^\top, U \right\rangle + |\widetilde{w}| \cdot (\mu(\widetilde{w}) - \mu(w))^\top U (\mu(\widetilde{w}) - \mu(w)) \\
&\overset{(b)}{\leq} (1 - \varepsilon) \langle M(\widetilde{w}), U \rangle + \|\mu(\widetilde{w}) - \mu(w)\|_2^2 \\
&\overset{(c)}{\leq} 2\gamma_2 + \|\mu(\widetilde{w}) - \mu(w)\|_2^2 \\
&\leq 2\gamma_2 + 2\|\mu(\widetilde{w}) - \mu\|_2^2 + 2\|\mu(w) - \mu\|_2^2 \\
&\overset{(d)}{\leq} 2\gamma_2 + 2\gamma_1^2 + 9(\gamma_2^2 + \varepsilon \|M(w)\|_2) < \frac{1}{60}\|M(w)\|_2 \\
&\leq \frac{1.1}{12}\lambda \leq \frac{1.1^2}{12} \sum_{i=1}^n w_i \tilde{\tau}_i \, , \quad\quad\quad\quad (13)
\end{aligned}
$$

for $\varepsilon$ sufficiently small and $\|M(w)\|_2 > \frac{100}{1.1}\gamma_2$. Here (a) follows since $w_i \leq \tilde{w}_i$ for $i \in S_g$, (b) follows since $\|U\|_2 \leq 1$, (c) follows from $\varepsilon$-goodness of $S_g$, and (d) follows from $\varepsilon$-goodness of $S_g$ and Lemma 6.2. Therefore overall we have $\sum_{i \in S_g} w_i \tilde{\tau}_i \leq \frac{1.1^3}{12} \sum_{i=1}^n w_i \tilde{\tau}_i$. Thus Theorem 5.2 applies, and $w'$ satisfies $\sum_{i=1}^n w_i' \tilde{\tau}_i \leq \frac{1}{4} \sum_{i=1}^n w_i \tilde{\tau}_i$ as well as $w \in \mathfrak{S}_{n,\varepsilon}$. Applying the guarantee that $\tilde{\tau}_i = (1 \pm 0.1)\tau_i$ again yields that

$$
\langle M(w'), U \rangle = \sum_{i=1}^n w_i' \tau_i \leq 1.1 \sum_{i=1}^n w_i' \tilde{\tau}_i \leq \frac{1.1}{4} \sum_{i=1}^n w_i \tilde{\tau}_i \leq \frac{1.1^2}{4} \sum_{i=1}^n w_i \tau_i = \frac{1.1^2}{4} \langle M(w), U \rangle \leq 0.31 \langle M(w), U \rangle \, .
$$

This completes the proof of the claim. $\quad\square$

Notice that Claim 6.7 immediately implies that the first invariant $w \in \mathfrak{S}_{n,\varepsilon}$ always holds. We now turn to proving the second invariant. Let $T$ be the number of iterations that the epoch runs for. Observe that for all $t = 0, \ldots, T$, we have that $M(w_t) \preceq M(w_0) \preceq \frac{1}{\alpha} I$, and so we are indeed in the setting of the guarantee in Section 5.4. Thus, by (8), since $F_t = M(w_{t+1})$, we obtain the following regret bound:

$$
\left\| \sum_{t=0}^{T-1} M(w_{t+1}) \right\|_2 \leq \sum_{t=0}^{T-1} \langle M(w_{t+1}), U_t \rangle + \alpha \sum_{t=0}^{T-1} \langle U_t, M(w_{t+1}) \rangle \, \|M(w_{t+1})\|_2 + \frac{\log d}{\alpha}
$$

$$
\leq 2 \sum_{t=0}^{T-1} \langle M(w_{t+1}), U_t \rangle + \|M(w_0)\|_2 \cdot \log d \, , \tag{14}
$$

where the second inequality follows by our choice of $\alpha$. We claim that for all $t > 0$, we must have $\langle M(w_t), U_t \rangle \leq 0.31 \|M(w_0)\|_2$. There are two cases. If we enter the if statement in Line 18, then

$$
\langle M_t, U_t \rangle = \sum_{i=1}^{n} w_i \tau_i \leq 1.1 \sum_{i=1}^{n} w_i \tilde{\tau}_i \leq \frac{1.1}{5} \lambda \leq 0.31 \|M(w_0)\|_2 \, ,
$$

and $M(w_{t+1}) = M(w_t)$, so this is clearly satisfied. Otherwise, the desired bound follows by Claim 6.7. Thus overall, by (14) we have that

$$
\left\| \sum_{t=0}^{T-1} M(w_t) \right\|_2 \leq T \cdot 0.62 \|M(w_0)\|_2 + \|M(w_0)\|_2 \cdot \log d \, . \tag{15}
$$

By Lemma 6.4, we further have that $M(w_{t+1}) \preceq M(w_t)$ for all $t = 0, \ldots, T-1$, and so this implies that

$$
T \|M(w_T)\|_2 \leq 0.62 \|M(w_0)\|_2 + \|M(w_0)\|_2 \cdot \log d \, .
$$

Simplifying both sides yields that if $T = C \log n$ for some sufficiently large constant $C$, then $\|M(w_T)\|_2 \leq \frac{2}{3} \|M(w_0)\|_2$. Thus after $O(\log d)$ iterations, we must terminate. $\qquad\square$

# 7 An MMW algorithm for robust mean estimation for sub-gaussian distributions

In this section we give an analog of the result in Section 6 but but in the setting where the distribution $D$ is subgaussian, and has identity covariance. We again first identify a deterministic condition for the inliers under which our algorithms will succeed. In this case, we need a stricter analog of $\varepsilon$-goodness. Specifically, we will require:

**Definition 7.1** (Subgaussian goodness). Let $D$ be a distribution with covariance Id and mean $\mu$. We say a set of points $S$ is $(\varepsilon, \gamma_1, \gamma_2)$-*subgaussian good* (or $(\varepsilon, \gamma_1, \gamma_2)$-s.g. good for short) with respect to $D$ if there exist universal constants $C_1, C_2$ so that the following inequalities are satisfied:

- $\|\mu(S) - \mu\| \leq \gamma_1$ and $\left\| \frac{1}{|S|} \sum_{i \in S} (X_i - \mu(S)) (X_i - \mu(S))^\top - \mathrm{Id} \right\|_2 \leq \gamma_2$, and

- For any subset $T \subset S$ so that $|T| = 2\varepsilon |S|$, we have

$$
\left\| \frac{1}{|T|} \sum_{i \in T} X_i - \mu \right\| \leq C_1 \cdot \sqrt{\log 1/\varepsilon} \, , \text{ and } \left\| \frac{1}{|T|} \sum_{i \in T} (X_i - \mu(S)) (X_i - \mu(S))^\top - \mathrm{Id} \right\|_2 \leq C_2 \cdot \log 1/\varepsilon \, .
$$

We have the following concentration inequality:

**Lemma 7.1** (see Lemmata 2.1.8 and 2.1.9 in [11])**.** *Let $X_1, \ldots, X_n \sim D$, where $D$ is subgaussian with variance proxy 1. Then, for any $\varepsilon > 0$ sufficiently small, we have that $S = \{X_1, \ldots, X_n\}$ is $(\varepsilon, \gamma_1, \gamma_2)$-s.g. good with probability $1 - \delta$, where*

$$\gamma_1 = O\left(\sqrt{\frac{d + \log 1/\delta}{n}}\right) , \ \text{and} \ \gamma_2 = O\left(\min \sqrt{\frac{d + \log 1/\delta}{n}}, \frac{d + \log 1/\delta}{n}\right) . \tag{16}$$

In particular, note that when $n = \Omega(\frac{d + \log 1/\delta}{\varepsilon^2 \log 1/\varepsilon})$, then Lemma 7.1 implies that $n$ i.i.d. samples from an isotropic sub-gaussian distribution is $(\varepsilon, O(\varepsilon\sqrt{\log 1/\varepsilon}), O(\varepsilon\sqrt{\log 1/\varepsilon}))$-s.g. good with probability $1 - \delta$. We will also require the following simple consequences of subgaussian goodness.

**Fact 7.2.** *Let $D$ be an isotropic distribution. Let $S$ be $(\varepsilon, \gamma_1, \gamma_2)$-s.g. good w.r.t. $D$. Then:*

- *for all $w' \in \Gamma_n$ with $w' \leq \frac{1}{n}\mathbf{1}_n$ and $|w'| \leq 2\varepsilon$, and for all unit vectors $v \in \mathbb{R}^d$, we have*

$$\sum_{i=1}^{n} w_i' \langle X_i - \mu, v \rangle^2 \leq O\left(\varepsilon \log 1/\varepsilon\right) .$$

- *if $w \in \Gamma_n$ satisfies $w \leq \frac{1}{n}\mathbf{1}_n$ and $\left|\frac{1}{n}\mathbf{1}_n - w\right| \leq 2\varepsilon$, then*

$$\left\|\sum_{i=1}^{n} w_i \left(X_i - \mu\right) \left(X_i - \mu\right)^\top - \mathrm{Id}\right\|_2 \leq \gamma_2 + \gamma_1^2 + O(\varepsilon \log 1/\varepsilon) , \ \text{and} \tag{17}$$

$$\left\|\sum_{i=1}^{n} w_i \left(X_i - \mu(w)\right) \left(X_i - \mu(w)\right)^\top - \mathrm{Id}\right\|_2 \leq \gamma_2 + 4\gamma_1^2 + O(\varepsilon \log 1/\varepsilon) . \tag{18}$$

*Proof.* We first prove the first claim. Let $w'' \in \Gamma_n$ be anything so that $w' \leq w'' \leq \frac{1}{n}\mathbf{1}_n$ and $|w''| = \varepsilon$. Then since all quantities on the LHS of the expression are nonnegative, we have that

$$\sum_{i=1}^{n} w_i' \langle X_i - \mu, v \rangle^2 \leq \sum_{i=1}^{n} w_i'' \langle X_i - \mu, v \rangle^2 . \tag{19}$$

Now let $A = \{w'' \in \Gamma_n : |w''| = \varepsilon\}$. This set is clearly convex, and moreover, by inspection, the vertices of $A$ are exactly given by $\frac{1}{n}\mathbf{1}_T$ where $|T| = 2\varepsilon n$. Thus, by convexity, the maximum of the RHS of (19) over $w'' \in A$ is obtained by $w'' = \frac{1}{n}\mathbf{1}_T$ for some $T$ with $|T| = 2\varepsilon n$. But then we have

$$\frac{1}{n} \sum_{i \in T} \langle X_i - \mu, v \rangle^2 = 2\varepsilon \cdot \frac{1}{|T|} \sum_{i \in T} \left(\langle X_i - \mu, v \rangle^2 - 1\right) + \varepsilon$$

$$\leq 2\varepsilon \left\|\frac{1}{|T|} \sum_{i \in T} \left(X_i - \mu\right) \left(X_i - \mu\right)^\top - \mathrm{Id}\right\|_2 + \varepsilon$$

$$\leq O(\varepsilon \log 1/\varepsilon) ,$$

by the s.g.-goodness of $S$. This completes the proof of the first bullet point.

We now turn our attention to the second claim. We have that

$$\left\|\sum_{i=1}^{n} w_i \left(X_i - \mu\right) \left(X_i - \mu\right)^\top - \mathrm{Id}\right\|_2 = \left\|\sum_{i=1}^{n} \frac{1}{n} \left(X_i - \mu\right) \left(X_i - \mu\right)^\top - \mathrm{Id} + \sum_{i=1}^{n} \left(\frac{1}{n} - w_i\right) \left(X_i - \mu\right) \left(X_i - \mu\right)^\top\right\|_2$$

$$\leq \left\|\sum_{i=1}^{n} \frac{1}{n} \left(X_i - \mu\right) \left(X_i - \mu\right)^\top - \mathrm{Id}\right\|_2 + O(\varepsilon \log 1/\varepsilon) ,$$

by the first claim. Further expanding, we have

$$\left\| \sum_{i=1}^{n} \frac{1}{n} \left(X_i - \mu\right)\left(X_i - \mu\right)^\top - \mathrm{Id} \right\|_2 = \left\| \sum_{i=1}^{n} \frac{1}{n} \left(X_i - \mu(S)\right)\left(X_i - \mu(S)\right)^\top - \mathrm{Id} + |w| \left(\mu(S) - \mu\right)\left(\mu(S) - \mu\right)^\top \right\|_2$$

$$\leq \left\| \sum_{i=1}^{n} \frac{1}{n} \left(X_i - \mu(S)\right)\left(X_i - \mu(S)\right)^\top - \mathrm{Id} \right\|_2 + \|\mu(S) - \mu\|_2^2$$

$$\leq \gamma_2 + \gamma_1^2 \, .$$

Putting it all together yields (17). To prove (18), simply observe that

$$\left\| \sum_{i=1}^{n} w_i \left(X_i - \mu\right)\left(X_i - \mu\right)^\top - \sum_{i=1}^{n} w_i \left(X_i - \mu(w)\right)\left(X_i - \mu(w)\right)^\top \right\|_2 = |w| \left\| \left(\mu(w) - \mu\right)\left(\mu(w) - \mu\right)^\top \right\|_2$$

$$\leq O(\gamma_1^2) + O(\epsilon \log(1/\epsilon)) \, .$$

In the last step we have used the definition of $(\epsilon, \gamma_1, \gamma_2)$-s.g. goodness and convexity. $\qquad \square$

As a result, we also have the following tail bound on mean shifts caused by small subsets of points:

**Corollary 7.3.** *Let $D$ be an isotropic distribution, and let Let $S$ be $(\varepsilon, \gamma_1, \gamma_2)$-s.g. good w.r.t. $D$. Then:*

- *for all $w' \in \Gamma_n$ with $w' \leq \frac{1}{n}\mathbf{1}_n$ and $|w'| \leq 2\varepsilon$, we have*

$$\left\| \sum_{i=1}^{n} w_i' \left(X_i - \mu\right) \right\|_2 \leq O\left(\varepsilon\sqrt{\log 1/\varepsilon}\right) \, , \quad and$$

- *if $w \in \Gamma_n$ satisfies $w \leq \frac{1}{n}\mathbf{1}_n$ and $\left|\frac{1}{n}\mathbf{1}_n - w\right| \leq 2\varepsilon$, then*

$$\|\mu(w) - \mu\|_2 \leq \frac{1}{1-\varepsilon}\left(\gamma_1 + O\left(\varepsilon\sqrt{\log 1/\varepsilon}\right)\right) \, .$$

*Proof.* We first prove the first claim. Fix any unit vector $v \in \mathbb{R}^d$. Then we have

$$\left(\sum_{i=1}^{n} w_i' \langle X_i - \mu, v\rangle\right)^2 \overset{(a)}{\leq} |w'| \sum_{i=1}^{n} w_i' \langle X_i - \mu, v\rangle^2$$

$$\overset{(b)}{\leq} O(\varepsilon^2 \log 1/\varepsilon) \, ,$$

where (a) follows from Cauchy-Schwarz, and (b) follows from Fact 7.2. By taking square roots and a supremum over all unit vectors $v$, we obtain the desired conclusion.

We now prove the second claim. We expand

$$|w| \, \|\mu(w) - \mu\|_2 = \left\| \sum_{i=1}^{n} w_i \left(X_i - \mu\right) \right\|_2$$

$$= \left\| \sum_{i=1}^{n} \frac{1}{n} \left(X_i - \mu\right) + \sum_{i=1}^{n} \left(\frac{1}{n} - w_i\right)\left(X_i - \mu\right) \right\|_2$$

$$\leq \gamma_1 + O(\varepsilon\sqrt{\log 1/\varepsilon}) \, ,$$

where the last line follows from subgaussian goodness, and applying the first claim with $w_i' = \frac{1}{n} - w_i$. $\qquad \square$

## 7.1 Mean deviations to moment bounds

As before, we will require a lemma which relates the mean shift caused by a small fraction of points to spectral deviations. However, because in this case we will assume that our data is subgaussian, we will be able to prove stronger statements, which will in turn allow us to achieve much better error. We first record the following simple fact, which states that if we have a set of weights that puts almost all of its mass on a good set, then the restriction of that set of weights to the good set satisfies the conditions of the lemmata proved in the above section.

**Fact 7.4.** Let $\varepsilon < 1/2$, and suppose $S = S_g \cup S_b \setminus S_r$, where $S_g$ is $(\varepsilon, \gamma_1, \gamma_2)$-s.g. good, and $|S_b|, |S_r| \leq \varepsilon|S|$. Let $w \in \mathfrak{S}_{n,\varepsilon}$. Then $w_g \leq \frac{1}{|S_g|}\mathbf{1}_{S_g}$ and $\left|\frac{1}{|S_g|}\mathbf{1}_{S_g} - w_g\right| \leq 2\varepsilon$.

We first show that, no matter what, the smallest eigenvalue of the empirical covariance we choose cannot be too small. Formally:

**Lemma 7.5.** Let $\gamma_1, \gamma_2 > 0$. Suppose $S = S_g \cup S_b \setminus S_r$, where $S_g$ is $(\varepsilon, \gamma_1, \gamma_2)$-s.g. good, and $|S_b|, |S_r| \leq \varepsilon|S|$. Let $w \in \mathfrak{S}_{n,\varepsilon}$. Then

$$\sum_{i \in S_g \cap S} w_i \left(X_i - \mu(w)\right)\left(X_i - \mu(w)\right)^\top \succeq (1 - \xi)\,\mathrm{Id} \,,$$

where $\xi = \gamma_2 + 2\gamma_1^2 + O(\varepsilon \log 1/\varepsilon)$.

*Proof.* Let $w' \in \mathfrak{S}_{n,\varepsilon}$ be defined by $w'_i = w_i$ if $i \in S_g \cap S$ and $w'_i = 0$ otherwise. By Lemma 6.4, we know that

$$
\begin{aligned}
\sum_{i \in S_g \cap S} w_i \left(X_i - \mu(w)\right)\left(X_i - \mu(w)\right)^\top &\succeq \sum_{i \in S_g \cap S} w_i \left(X_i - \mu(w')\right)\left(X_i - \mu(w')\right)^\top \\
&\succeq \left(1 - \left(\gamma_2 + 2\gamma_1^2 + O(\varepsilon \log 1/\varepsilon)\right)\right)\mathrm{Id} \,,
\end{aligned}
$$

by 18 of Fact 7.2. $\qquad\square$

We now show the following:

**Lemma 7.6.** Let $\gamma_1, \gamma_2 > 0$. Suppose $S = S_g \cup S_b \setminus S_r$, where $S_g$ is $(\varepsilon, \gamma_1, \gamma_2)$-s.g. good, and $|S_b|, |S_r| \leq \varepsilon|S|$. Let $w \in \mathfrak{S}_{n,\varepsilon}$, and let $\lambda = \|M(w_t) - \mathrm{Id}\|_2$. Then

$$\|\mu(w) - \mu\|_2 \leq \frac{1}{1 - \varepsilon} \cdot \left(2\gamma_1 + \sqrt{\varepsilon(\lambda + \gamma_2)} + O(\varepsilon\sqrt{\log 1/\varepsilon})\right) \,.$$

Before we prove this lemma, observe that if $\gamma = O(\varepsilon\sqrt{\log 1/\varepsilon})$ and $\gamma_2 = O(\varepsilon \log 1/\varepsilon)$ and $\varepsilon \leq 1/2$ then the RHS of the lemma simplifies to $O(\varepsilon\sqrt{\log 1/\varepsilon})$.

*Proof of Lemma 7.6.* Let $\rho = \mu(w) - \mu$. As before, we have the following sequence of identities:

$$
\begin{aligned}
|w| \cdot \|\mu(w) - \mu\|_2^2 &= |w| \cdot \langle \mu(w) - \mu, \rho \rangle \\
&= \sum_{i=1}^n w_i \langle X_i - \mu, \rho \rangle \\
&= \underbrace{\sum_{i \in S_g \cap S} w_i \langle X_i - \mu, \rho \rangle}_{W_0} + \underbrace{\sum_{i \in S_b} w_i \langle X_i - \mu, \rho \rangle}_{W_1} \,.
\end{aligned}
$$

We treat the two terms on the RHS separately. We first consider $W_0$. We continue expanding, and observe:

$$\left| \sum_{i \in S_g \cap S} w_i \langle X_i - \mu, \rho \rangle \right| = \left| \sum_{i \in S_g \cap S} \frac{1}{n} \langle X_i - \mu, \rho \rangle + \sum_{i \in S_g \cap S} \left( \frac{1}{n} - w_i \right) \langle X_i - \mu, \rho \rangle \right|$$

$$\leq \left| \sum_{i \in S_g} \frac{1}{n} \langle X_i - \mu, \rho \rangle \right| + \left| \sum_{i \in S_r} \frac{1}{n} \langle X_i - \mu, \rho \rangle \right| + \left| \sum_{i \in S_g \cap S} \left( \frac{1}{n} - w_i \right) \langle X_i - \mu, \rho \rangle \right|$$

$$\overset{(a)}{\leq} \left| \sum_{i \in S_g} \frac{1}{n} \langle X_i - \mu, \rho \rangle \right| + O\left( \varepsilon \sqrt{\log 1/\varepsilon} \right) \|\rho\|_2$$

$$\overset{(b)}{\leq} \left( \gamma_1 + O\left( \varepsilon \sqrt{\log 1/\varepsilon} \right) \right) \|\rho\|_2 \ , \tag{20}$$

where (a) follows from two applications of Corollary 7.3, and (b) follows from Cauchy-Schwarz and subgaussian goodness.

We now turn our attention to bounding $W_1$. We have

$$|W_1| \leq \left| \sum_{i \in S_b} w_i \langle X_i - \mu(w), \rho \rangle \right| + \sum_{i \in S_b} w_i \|\rho\|_2^2$$

$$\leq \left| \sum_{i \in S_b} w_i \langle X_i - \mu(w), \rho \rangle \right| + \varepsilon \|\rho\|_2^2 \ .$$

Focusing in on the first term in the RHS, we have

$$\left( \sum_{i \in S_b} w_i \langle X_i - \mu(w), \rho \rangle \right)^2 \overset{(a)}{\leq} \left( \sum_{i \in S_b} w_i \right) \sum_{i \in S_b} w_i \langle X_i - \mu(w), \rho \rangle^2$$

$$= \left( \sum_{i \in S_b} w_i \right) \sum_{i \in S_b} w_i \left( \langle X_i - \mu(w), \rho \rangle^2 - \|\rho\|_2^2 \right) + \left( \sum_{i \in S_b} w_i \right)^2 \|\rho\|_2^2$$

$$\overset{(b)}{\leq} \varepsilon \sum_{i \in S_b} w_i \left( \langle X_i - \mu(w), \rho \rangle^2 - 1 \right) + \varepsilon^2 \|\rho\|_2^2 \ , \tag{21}$$

where (a) follows from Cauchy-Schwarz, and (b) follows since $w$ places at most $\varepsilon$ mass on $S_b$. Now observe that

$$\left| \sum_{i \in S_b} w_i \left( \langle X_i - \mu(w), \rho \rangle^2 \right) \right| = \left| \sum_{i=1}^{n} w_i \left( \langle X_i - \mu(w), \rho \rangle^2 \right) - \sum_{i \in S_g \cap S} w_i \left( \langle X_i - \mu(w), \rho \rangle^2 \right) \right|$$

$$\leq \|M(w_t) - \mathrm{Id}\|_2 \|\rho\|_2^2 + \left| \sum_{i \in S_g \cap S} w_i \left( \langle X_i - \mu(w), \rho \rangle^2 - \|\rho\|_2^2 \right) \right|$$

$$\leq \|M(w_t) - \mathrm{Id}\|_2 \|\rho\|_2^2 + \left\| \sum_{i \in S_g \setminus S_r} w_i \left( X_i - \mu(w) \right) \left( X_i - \mu(w) \right)^\top - \mathrm{Id} \right\|_2 \|\rho\|_2^2$$

$$\leq \left( \|M(w_t) - \mathrm{Id}\|_2 + \gamma_2 + 2\gamma_1^2 + O(\varepsilon \log 1/\varepsilon) \right) \|\rho\|_2^2 \ ,$$

by Fact 7.2, where we take the convention that $w_i = 0$ for $i \in S_r$. Hence, combining terms, recalling the definition of $\lambda = \|M(w_t) - \mathrm{Id}\|_2$, and taking square roots, we have

$$|W_1| \leq \left( \sqrt{\varepsilon(\lambda + \gamma_2)} + \gamma_1\sqrt{2\varepsilon} + O\left( \varepsilon \sqrt{\log 1/\varepsilon} \right) \right) \cdot \|\rho\|_2 + \varepsilon \|\rho\|_2^2 \ .$$

Combining this with (20) yields

$$\|\rho\|_2^2 \leq |W_0| + |W_1|$$
$$\leq \left(2\gamma_1 + \sqrt{\varepsilon(\lambda + \gamma_2)} + O(\varepsilon\sqrt{\log 1/\varepsilon})\right)\|\rho\|_2 + \varepsilon\|\rho\|_2^2 .$$

Solving for $\|\rho\|_2$ yields the desired claim. $\qquad\square$

## 7.2 Algorithm description

The algorithm is quite similar to the algorithm presented in Section 6 for the bounded covariance case. The formal pseudocode is presented in Algorithm 4. For any $w \in \Gamma_n$, let $\mu(w)$ and $M(w)$ be as in Section 6. However, we will require a slightly stronger notion of score oracle than before.

Recall that before, given a dataset $S$, and a sequence of weights $w_0, \ldots, w_{t-1}$, the score oracle is asked to produce multiplicative approximations to $\tau_{t,i}$ where $\tau_{t,i}$ is defined as (10). One consequence of this is that this allows us to produce multiplicative approximations to $\left\langle M(w_t^{(s)}), U_t^{(s)} \right\rangle = \sum_{i=1}^n w_{t,i}\tau_{t,i}$. However, we will require multiplicative approximations to $\left\langle M(w_t^{(s)}) - \mathrm{Id}, U_t^{(s)} \right\rangle$, which cannot be obtained black-box via multiplicative approximations to the original scores.

To rectify this, we say that an algorithm $\mathcal{O}^*$ is an *augumented score oracle* if it takes a dataset $S$ and a sequence of weights $w_0, \ldots, w_{t-1}$, and outputs $\tilde{\tau}_{t,i}$ for all $i = 1, \ldots, n$, but also an overall score $q_t$ which is intended to approximate $\left\langle M(w_t^{(s)}) - \mathrm{Id}, U_t^{(s)} \right\rangle$.

Given $S$ and such an oracle $\mathcal{O}^*$, the algorithm again proceeds in epochs. Initially, we let $w^{(0)} = \frac{1}{n}\mathbf{1}_n$. In epoch $s = 0, \ldots, L-1$, we proceed as follows. First, compute $\lambda^{(s)} \approx_{0.1} \left\|M(w^{(s)}) - \mathrm{Id}\right\|_2$. If $\lambda^{(s)} \leq O\left(\gamma_2 + \gamma_1 + \gamma_1^2 + \varepsilon\log 1/\varepsilon\right)$, we terminate and output $\mu(w^{(s)})$.

Otherwise, we let $w_0^{(s)} = w^s$. Then, in iteration $t = 0, \ldots, T_s - 1$, we first (approximately) compute $\lambda_t^{(s)} \approx_{0.1} \|M(w_t^{(s)}) - \mathrm{Id}\|_2$. If $\lambda_t^{(s)} \leq \frac{1}{2}\lambda_0^{(s)}$, we terminate and let $w^{(s+1)} = w_t^{(s)}$. Otherwise, we let $U_t^{(s)}$ be prescribed by the MMW update with parameter $\alpha = 1/(1.1 \cdot \lambda^{(()s)})$. Then, produce the gain matrix is given as follows.

At time $t$, run $\mathcal{O}^*$ given $S$ and the sequence of weights $w_0^{(s)}, \ldots, w_{t-1}^{(s)}$ to obtain scores $\tilde{\tau}_{t,i}^{(s)}$ as well as an overall score $\tilde{q}_t^{(s)}$. Then, check if $\tilde{q}_t^{(s)} \leq \frac{1}{5}\lambda_0^{(s)}$. If so, then we let $w_{t+1}^{(s)} = w_t^{(s)}$.

Otherwise, sort the $\tilde{\tau}_{t,i}^{(s)}$ in descending order. WLOG assume that $\tilde{\tau}_{t,1}^{(s)} \geq \tilde{\tau}_{t,2}^{(s)} \geq \ldots \geq \tilde{\tau}_{t,n}^{(s)}$. Let $m$ be the smallest integer so that $\sum_{i \leq m} w_{t,i}^{(s)} \geq 2\varepsilon$. Then, run 1DFILTER on $\tilde{\tau}_{t,1}^{(s)}, \ldots, \tilde{\tau}_{t,m}^{(s)}$ with corresponding weights $w_{t,1}^{(s)}, \ldots, w_{t,m}^{(s)}$, to obtain a new set of weights $w_1', \ldots, w_m'$, and let $w_{t+1,i}^{(s)}$ be defined by

$$w_{t+1,i}^{(s)} = \begin{cases} w_{t,i}^{(s)} & \text{if } i > m; \\ w_i' & \text{if } i \leq m. \end{cases} \tag{22}$$

That is, we find the largest $2\varepsilon$-percentile of the scores weighted by the current weights, and run the univariate filter on these set of weights, leaving the other weights unchanged. Finally, we output the gain matrix $F_t^{(s)} = M(w_{t+1}^{(s)}) - \mathrm{Id}$. Notice that this matrix may not be PSD.

**The score oracles** The exact score oracle $\mathcal{O}^*_{\text{exact}}$ would, given $S$, and given $w_0^{(s)}, \ldots, w_{t-1}^{(s)}$, would output scores $\tau_{t,i}^{(s)}$ given by (10) with $U_t^{(s)}$ as given by Algorithm 4. The exact score oracle would also output

$$q_t^{(s)} = \left\langle M(w_t^{(s)}) - \mathrm{Id}, U_t^{(s)} \right\rangle . \tag{23}$$

Observe that the fact that the $F_t^{(s)}$ includes a negative identity term does not affect these scores at all, and indeed we can take $U_t^{(s)}$ to be as in (11), since the parameter $c$ is chosen in any case to normalize $U_t^{(s)}$ to have trace 1.

---

**Algorithm 4** MMW-based filtering method for robust mean estimation for subgaussian distributions

---

1: **Input:** dataset $S \subset \mathbb{R}^d$ of size $n$, parameters $\gamma_1, \gamma_2$, augmented score oracle $\mathcal{O}$
2: Let $C > 0$ be a sufficiently large universal constant.
3: Let $w^{(0)} = \frac{1}{n}\mathbf{1}_n$.
4: **for** epoch $s = 0, \ldots, O(\log \kappa)$ **do**
5:      Let $\lambda^{(s)} \approx_{0.1} \left\| M(w^{(s)}) - \mathrm{Id} \right\|_2$
6:      **if** $\lambda^{(s)} \leq C \cdot \left( \gamma_2 + \gamma_1 + \gamma_1^2 + \varepsilon \log 1/\varepsilon \right)$ **then**
7:          **return** $\mu(w^{(s)})$
8:      **end if**
9:      Let $w_0^{(s)} = w^{(s)}$
10:      Let $\alpha^{(s)} = \frac{1}{1.1 \cdot \lambda_0^{(s)}}$
11:      **for** iteration $t = 0, \ldots, O(\log d)$ **do**
12:          Compute $\lambda_t^{(s)} \approx_{0.1} \| M(w_t^{(s)}) - \mathrm{Id} \|_2$.
13:          **if** $\lambda_t^{(s)} \leq \frac{1}{2} \lambda_0^{(s)}$ **then**
14:             terminate epoch
15:          **end if**
16:          Let $U_t^{(s)}$ be given by MMW update with parameter $\alpha^{(s)}$
17:          For $i = 1, \ldots, n$, let $\tilde{\tau}_{t,i}^{(s)}, \tilde{q}_t^{(s)} = \mathcal{O}(S, w_0^{(s)}, \ldots, w_n^{(s)})$
18:          **if** $\tilde{q}_t^{(s)} \leq \frac{1}{1.1 \cdot 5} \lambda_0^{(s)}$ **then**
19:             Let $w_{t+1}^{(s)} = w_t^{(s)}$.
20:          **else**
21:             Sort the $\tilde{\tau}_{t,i}^{(s)}$ in descending order.
22:             WLOG assume that $\tilde{\tau}_{t,1}^{(s)} \geq \tilde{\tau}_{t,2}^{(s)} \geq \ldots \geq \tilde{\tau}_{t,n}^{(s)}$
23:             Let $m$ be the smallest integer so that $\sum_{i \leq m} w_{t,i}^{(s)} \geq 2\varepsilon$.
24:             Let $w' = 1\mathrm{DF}\textsc{ilter}((w_{t,1}^{(s)}, \ldots, w_{t,m}^{(s)}), (\tilde{\tau}_{t,1}^{(s)}, \ldots, \tilde{\tau}_{t,m}^{(s)}), 1/4)$.
25:             Let $w_{t+1}^{(s)}$ be as defined in (22).
26:          **end if**
27:          Let feedback matrix to MMW update be $F_t = M(w_{t+1}^{(s)})$
28:      **end for**
29:      Let $w^{(s+1)} = w_t^{(s)}$
30: **end for**

---

As before, we cannot access these exact scores in nearly-linear time, so instead we ask for approximations. Specifically, we will assume an approximate augmented score oracle $\mathcal{O}_{\mathrm{approx}}^*$, which given $S$ and $w_0^{(s)}, \ldots, w_{t-1}^{(s)}$, output scores $\tilde{\tau}_{t,i}^{(s)}$ so that $\tilde{\tau}_{t,i}^{(s)} \approx_{0.1} \tau_{t,i}^{(s)}$ for all $i = 1, \ldots, n$, as well as $\tilde{q}_t^{(s)}$ satisfying

$$\left| \tilde{q}_t^{(s)} - q_t \right| \leq 0.1 \cdot q_t + 0.05 \cdot \| M(w_t) - \mathrm{Id} \|_2 \ .$$

As before, the choice of constants here is arbitrary, and any constants sufficiently small will work. In Section 8 we construct such an approximate augmented score oracle in nearly-linear time.

## 7.3 Correctness of s.g.-QUEScoreFilter

The rest of this section is dedicated to the proof of the following theorem:

**Theorem 7.7.** *Let $D$ be a subgaussian isotropic distribution on $\mathbb{R}^d$ with mean $\mu$. Let $\varepsilon < c$, where $c$ is a universal constant, and let $\gamma_1, \gamma_2 > 0$. Let $S$ be a dataset, $|S| = n$, so that $S = S_g \cup S_b \setminus S_r$ so that $S_g$ is $(\varepsilon, \gamma_1, \gamma_2)$-s.g. good with respect to $D$, and $|S_b|, |S_r| \leq \varepsilon |S|$. Suppose that $\|X_i\|_2 \leq \kappa_1$ for all $i = 1, \ldots, n$,*

*and moreover* $\left\|M(\frac{1}{n}\mathbf{1}_n) - \mathrm{Id}\right\| \leq O(\varepsilon \log 1/\varepsilon + \varepsilon \kappa_2)$. *Then* s.g.-QUESCOREFILTER$(S, \varepsilon, \mathscr{O}_{\mathrm{approx}})$ *terminates after at most* $O(\log \kappa_2)$ *epochs, and outputs a* $w \in \mathfrak{S}_{n,\varepsilon}$ *so that*

$$\|\mu(w) - \mu\|_2 \leq O\left(\gamma_1 + \varepsilon\sqrt{\log 1/\varepsilon} + \sqrt{\varepsilon(\gamma_1 + \gamma_2)}\right) .$$

*Moreover, each epoch runs for at most* $O(\log d)$ *iterations, requires* $O(\log d)$ *calls to* $\mathscr{O}_{\mathrm{approx}}$, *and requires* $\widetilde{O}(nd + n\log \kappa_1)$ *additional computation.*

Our main lemma is the following:

**Lemma 7.8.** *The following invariants always hold. There exists some universal constant* $C > 0$ *so that for all epochs* $s$, *we have:*

- $w^{(s)} \in \mathfrak{S}_{n,\varepsilon}$, *and*

- *If*

$$\left\|M(w^{(s)}) - \mathrm{Id}\right\|_2 > C \cdot \left(\gamma_2 + \gamma_1 + \gamma_1^2 + \varepsilon \log 1/\varepsilon\right) ,$$

  *then epoch* $s$ *terminates after* $O(\log d)$ *iterations, and outputs* $w^{(s+1)}$ *so that* $\left\|M(w^{(s)}) - \mathrm{Id}\right\|_2 \leq \frac{3}{4}\left\|M(w^{(s)}) - \mathrm{Id}\right\|_2$.

We first demonstrate how this lemma proves Theorem 7.7.

*Proof of Theorem 7.7 given Lemma 7.8.* By our condition on $M(\frac{1}{n}\mathbf{1}_n)$, after at most $s = O(\log \kappa_2)$ iterations, we must have that

$$\left\|M(w^{(s)}) - \mathrm{Id}\right\|_2 \leq C \cdot \left(\gamma_2 + \gamma_1 + \gamma_1^2 + \varepsilon \log 1/\varepsilon\right) .$$

Since $w^{(s)} \in \mathfrak{S}_{n,\varepsilon}$, Lemma 7.6 implies that for $\varepsilon < c$ sufficiently small, we have

$$\|\mu(w) - \mu\|_2 \leq O\left(\gamma_1 + \varepsilon\sqrt{\log 1/\varepsilon} + \sqrt{\varepsilon(\gamma_1 + \gamma_2)}\right) ,$$

as claimed.

We now turn to bounding the runtime. As in Theorem 6.5, it is clear that we make at most $\log d$ calls to the oracle every epoch, and 1DFILTER runs in time $O(n\log \kappa_1)$. Moreover, the approximate eigenvalue computations can still be done in $\widetilde{O}(nd)$ time since we may run power method on $M(w_s^{(t)}) - \mathrm{Id}$, as we can evaluate matrix-vector multiplications against this matrix in $O(nd)$ time. This completes the proof. $\qquad\square$

The remainder of the section is dedicated to the proof of this lemma. As in the previous section, for simplicity of notation, as we will only consider a fixed epoch $s$, we will drop the superscripts.

The proof of Lemma 7.8 breaks down into two parts. First, we will show that assuming we have not yet made sufficient progress, we remain in the regime where the filter is guaranteed to make progress, i.e., the majority of the mass of the $\tau_i$ are from bad points. This is captured in the following lemma:

**Lemma 7.9.** *At time* $t$, *suppose that* $\langle M_t - \mathrm{Id}, U_t \rangle > \frac{1}{1.1 \cdot 5}\lambda_0$. *Then* $w_{t+1} \in \mathfrak{S}_{n,\varepsilon}$ *and* $\langle F_t, U_t \rangle \leq \frac{1}{4}\langle F_{t-1}, U_t \rangle$.

Then, we will show that this implies that the regret bounds of MMW guarantee that we make constant progress in logarithmically many iterations:

**Lemma 7.10.** *Suppose for all* $t = 0, \ldots, T-1$, *Lemma 7.9 holds, where* $T = O(\log d)$. *Then,* $\|M(w_T) - \mathrm{Id}\|_2 \leq 0.63 \cdot \|M(w_0) - \mathrm{Id}\|$.

*Proof of Lemma 7.9.* Recall that $m$ was chosen to be the $2\varepsilon$-percentile of the scores under the weighting given by $w$, and as before, for simplicity assume that the $\tilde{\tau}_{t,i}$ are in decending order. The main work in this proof will be to show that the scores $\tilde{\tau}_{t,1}, \dots, \tilde{\tau}_{t,m}$ and weights $w_{t,1}, \dots, w_{t,m}$ satisfy the conditions of Theorem 5.2.

The condition that $\langle M_t - \mathrm{Id}, U_t \rangle > \frac{1}{1.1\cdot5}\lambda_0$ implies that in this case, we will run the univariate filter. Let $S'_g = S_g \cap [m]$ and let $S'_b = S_b \cap [m]$, and let $w'_g$ and $w'_b$ the restriction of $w_t$ to $S'_g$ and $S'_b$, respectively. Observe that $\sum_{i \in S'_b} w_i \le \sum_{i \in S_b} w_i \le \varepsilon$, and therefore $\sum_{i \in S'_g} w_i \in [\varepsilon, 2\varepsilon]$. We then have

$$\sum_{i \in S'_g} w_{t,i}\tau_{t,i} = \sum_{i \in S'_g} w_{t,i} \left\langle (X_i - \mu(w_t))(X_i - \mu(w_t))^\top, U_t \right\rangle$$

$$\overset{(a)}{\le} 2\sum_{i \in S'_g} w_{t,i} \left\langle (X_i - \mu)(X_i - \mu)^\top, U_t \right\rangle + 2|w'_g| \left\langle (\mu - \mu(w_t))(\mu - \mu(w_t))^\top, U_t \right\rangle$$

$$\overset{(b)}{\le} O(\varepsilon \log 1/\varepsilon) + 2|w'_g| \left\langle (\mu - \mu(w_t))(\mu - \mu(w_t))^\top, U_t \right\rangle$$

$$\le O(\varepsilon \log 1/\varepsilon) + 4\varepsilon \|\mu - \mu(w_t)\|_2^2$$

$$\overset{(c)}{\le} O(\varepsilon \log 1/\varepsilon) + 4\varepsilon \left( \gamma_1 + O(\varepsilon\sqrt{\log 1/\varepsilon}) + \sqrt{\varepsilon(\lambda_t + \gamma_2)} \right)^2$$

$$\le O(\varepsilon \log 1/\varepsilon) + 8\varepsilon\gamma_1 + 8\varepsilon^2(\lambda_t + \gamma_2)$$

$$\le \frac{1}{30} \langle M(w_t) - \mathrm{Id}, U_t \rangle \ . \tag{24}$$

where (a) follows since for any vectors $x, y, z$, we have $(x-y)(x-y)^\top \preceq 2(x-z)(x-z)^\top + 2(y-z)(y-z)^\top$, (b) follows from Fact 7.2, (c) follows from Lemma 7.6, and the last line follows from our assumption on $\langle M(w_t) - \mathrm{Id}, U_t \rangle$. From this we conclude that $\sum_{i \in S'_g} w_{t,i}\tilde{\tau}_{t,i} \le \frac{1}{29} \langle M(w_t) - \mathrm{Id}, U_t \rangle$.

Note that as a consequence of this, we have that $\tilde{\tau}_{t,m} \le O(\log 1/\varepsilon) + 8\varepsilon\gamma_1 + 8\varepsilon(\lambda_t + \gamma_2)$, since that is an upper bound on the average value of the $\tilde{\tau}_{t,i}$ for $i \in S'_g$, as $|w'_g| \ge \varepsilon$.

We now turn to lower bound the contribution from $S'_b$. First observe that

$$\sum_{i \in S_b} w_i \tau_{t,i} = \sum_{i \in S_b} w_{t,i} \left\langle (X_i - \mu(w_t))(X_i - \mu(w_t))^\top, U_t \right\rangle$$

$$= \sum_{i \in S_b} w_{t,i} \left\langle (X_i - \mu(w_t))(X_i - \mu(w_t))^\top - \mathrm{Id}, U_t \right\rangle - |w_b|$$

$$= \langle M(w_t) - \mathrm{Id}, U_t \rangle - \sum_{i \in S_g \cap S} w_{t,i} \left\langle (X_i - \mu(w_t))(X_i - \mu(w_t))^\top - \mathrm{Id}, U_t \right\rangle - |w_b|$$

$$\overset{(a)}{\ge} \langle M(w_t) - \mathrm{Id}, U_t \rangle - \left( \gamma_2 + 2\gamma_1^2 + O(\varepsilon \log 1/\varepsilon) \right)$$

$$\ge \frac{99}{100} \langle M(w_t) - \mathrm{Id}, U_t \rangle \ ,$$

where (a) follows from Fact 7.2 and since $|w_b| \le \varepsilon$, and the last inequality follows by our assumption on

$\langle M(w_t) - \mathrm{Id}, U_t \rangle$. Therefore

$$\sum_{i \in S'_b} w_i \tau_{t,i} = \sum_{i \in S_b} w_i \tau_{t,i} - \sum_{i \in S_b \setminus S'_b} w_i \tau_{t,i}$$

$$\geq \frac{99}{100} \langle M(w_t) - \mathrm{Id}, U_t \rangle - \sum_{i \in S_b \setminus S'_b} w_i \tau_{t,i}$$

$$\overset{(a)}{\geq} \frac{99}{100} \langle M(w_t) - \mathrm{Id}, U_t \rangle - 1.21 \cdot |w_b| \left( O(\log 1/\varepsilon) + 8\varepsilon \gamma_1 + 8\varepsilon(\lambda_t + \gamma_2) \right)$$

$$\overset{(b)}{\geq} \frac{49}{50} \langle M(w_t) - \mathrm{Id}, U_t \rangle \ , \tag{25}$$

where (a) follows since $\tau_{t,i} \leq 1.1 \tilde{\tau}_{t,i} \leq \cdot 1.1^2 \cdot \tau_{t,m}$ for all $i \in S_b \setminus S'_b$, and (b) follows from our assumption on $\langle M(w_t) - \mathrm{Id}, U_t \rangle$.

Equations (24) and (25) together imply that our set of scores in this instance will be in the setting of Theorem 5.2. Thus Theorem 5.2 guarantees that the univariate filter outputs a set of weights $w'_1, \ldots, w'_m$ so that $\sum_{i=1}^m w'_i \tilde{\tau}_{t,i} \leq \frac{1}{5} \sum_{i=1}^m w_{t,i} \tilde{\tau}_{t,i}$. Notice that in particular this implies that $w_{t+1} \in \mathfrak{S}_{n,\varepsilon}$, which proves one of the claims in the Lemma. To complete the proof of the lemma, observe that

$$\langle M(w_t) - \mathrm{Id}, U_t \rangle - \langle F_t, U_t \rangle = \sum_{i=1}^m (w_{t,i} - w'_t)(\tau_{t,i} - 1)$$

$$\overset{(a)}{\geq} \sum_{i=1}^m (w_{t,i} - w'_t)\tau_{t,i} - 2\varepsilon$$

$$\overset{(b)}{\geq} \frac{4}{1.21 \cdot 5} \sum_{i=1}^m w_{t,i} \tau_{t,i} - 2\varepsilon$$

$$\overset{(c)}{\geq} \frac{4}{1.21 \cdot 5} \frac{49}{50} \langle M(w_t) - \mathrm{Id}, U_t \rangle - 2\varepsilon$$

$$\overset{(d)}{\geq} 0.63 \langle M(w_t) - \mathrm{Id}, U_t \rangle \ ,$$

where (a) follows since $m$ is the $2\varepsilon$-percentile, (b) follows from the guarantee of Theorem 5.2, (c) follows from (25), and (d) follows from our assumption on $\lambda_t$. Rearranging terms completes the proof. $\qquad\square$

We now show that this is enough to guarantee Lemma 7.10, which guarantees we make constant multiplicative progress in every epoch.

*Proof of Lemma 7.10.* Observe that if we terminate prematurely we clearly satisfy the lemma. Thus we may assume we do not terminate until timestep $T - 1$. Lemma 7.9 then implies that no matter which update we do at time $t$ for $t = 0, \ldots, T - 1$, we have the guarantee that

$$\langle F_t, U_t \rangle \leq 0.63 \langle F_{t-1}, U_t \rangle \leq 0.63 \langle F_0, U_t \rangle \leq 0.63 \|M(w_0) - \mathrm{Id}\|_2 \ . \tag{26}$$

Moreover, by Lemma 6.4, we have that $M(w_t) - \mathrm{Id} \preceq M(w_0) - \mathrm{Id}$, and hence $\frac{1}{\alpha}(M(w_t) - \mathrm{Id}) \preceq I$ by our choice of $\alpha$. Therefore by our regret bound, we have

$$\left\| \sum_{i=0}^{T-1} (M(w_t) - \mathrm{Id}) \right\|_2 \leq \sum_{t=0}^{T-1} \langle U_t, M(w_t) - \mathrm{Id} \rangle + \sum_{u=0}^{T-1} \langle U_t, |M(w_t) - \mathrm{Id}| \rangle \frac{\|M(w_t) - \mathrm{Id}\|_2}{\|M(w_0) - \mathrm{Id}\|_2} + \log(n) \cdot \|M(w_0) - \mathrm{Id}\|_2 \ . \tag{27}$$

By Lemma 7.5, we know that for all $t = 0, \ldots, T - 1$, we must have

$$M(w_t) - \mathrm{Id} \succeq -\left( \gamma_2 + 2\gamma_1^2 + O(\varepsilon \log 1/\varepsilon) \right) \mathrm{Id} \ . \tag{28}$$

This will allow us to simplify a number of expressions. In particular, since $M(w_t) - \mathrm{Id} \preceq M(w_0) - \mathrm{Id}$, this implies that all positive eigenvalues of $M(w_t) - \mathrm{Id}$ are smaller than $\lambda_0$, and (28) implies that all negative eigenvalues are bounded in absolute value by $\lambda_0$, by our assumption on $\lambda_0$. Hence

$$\frac{\|M(w_t) - \mathrm{Id}\|_2}{\|M(w_0) - \mathrm{Id}\|_2} \leq 1 \; . \tag{29}$$

Another implication is that

$$|M(w_t) - \mathrm{Id}| \preceq M(w_t) - \mathrm{Id} + 2\left(\gamma_2 + 2\gamma_1^2 + O(\varepsilon \log 1/\varepsilon)\right)\mathrm{Id} \; ,$$

and hence

$$\langle U_t, |M(w_t) - \mathrm{Id}|\rangle \leq \langle U_t, M(w_t) - \mathrm{Id}\rangle + 2\left(\gamma_2 + 2\gamma_1^2 + O(\varepsilon \log 1/\varepsilon)\right) \; . \tag{30}$$

Finally, we observe that (28) and Lemma 6.4 together imply that either

$$\|M(w_T) - \mathrm{Id}\|_2 \leq \gamma_2 + 2\gamma_1^2 + O(\varepsilon \log 1/\varepsilon) \quad \text{or} \tag{31}$$

$$\left\|\sum_{i=0}^{T-1}\left(M(w_t) - \mathrm{Id}\right)\right\|_2 \geq T \cdot \|M(w_{T-1}) - \mathrm{Id}\|_2 \; . \tag{32}$$

In the case of (31), we are clearly done, so we may assume that we are in the case of (32). Thus, plugging in (29), (30), and (32) into (27), and dividing by $T$, we obtain

$$\|M(w_{T-1}) - \mathrm{Id}\|_2 \leq \frac{2}{T}\sum_{i=0}^{T-1}\left(\langle U_t, M(w_t) - \mathrm{Id}\rangle\right) + 2\left(\gamma_2 + 2\gamma_1^2 + O(\varepsilon \log 1/\varepsilon)\right) + \frac{\log n}{T}\|M(w_0) - \mathrm{Id}\|_2$$

$$\overset{(a)}{\leq} \left(\frac{1}{2} + \frac{\log n}{T}\right)\|M(w_0) - \mathrm{Id}\| + 2 + 2\left(\gamma_2 + 2\gamma_1^2 + O(\varepsilon \log 1/\varepsilon)\right)$$

$$\overset{(b)}{\leq} \frac{3}{4}\|M(w_0) - \mathrm{Id}\|_2 \; ,$$

where (a) follows from (26), and (b) follows since $T = \Theta(\log d)$, and since $\|M(w_0) - \mathrm{Id}\|_2$ is a large constant factor larger than $\gamma_2 + 2\gamma_1^2 + O(\varepsilon \log 1/\varepsilon)$, by assumption. This completes the proof. $\square$

# 8 Fast approximate score oracles

In this section we describe how to implement the approximate score oracles and approximate augmented score oracles in nearly-linear time. Recall that an approximate score oracle takes as input a set of points $S \subset \mathbb{R}^d$ of size $n$, and a sequence of weights $w_0, \ldots, w_{t-1}, w_t$, and computes $\tilde{\tau}_t \in \mathbb{R}^n$, where for all $i = 1, \ldots, n$ we have $\tilde{\tau}_{t,i} \approx_{0.1} \tau_{t,i}$, where

$$\tau_{t,i} = (X_i - \mu(w_t))^\top U_t (X_i - \mu(w_t))^\top \; , \tag{33}$$

where

$$U_t = \exp\left(c\,\mathrm{Id} - \alpha\sum_{i=0}^{t-1}M(w_i)\right) = \frac{\exp\left(-\alpha\sum_{i=0}^{t-1}M(w_i)\right)}{\mathrm{tr}\exp\left(-\alpha\sum_{i=0}^{t-1}M(w_i)\right)} \; ,$$

and $\alpha > 0$ is a parameter. Additionally, recall that in Section 7 we additionally require that the score oracle be able to produce $\tilde{q}_t$ so that

$$|\tilde{q}_t - q_t| \leq 0.1 \cdot q_t + 0.05 \cdot \|M(w_t) - \mathrm{Id}\|_2 \; , \text{where } q_t = \langle M(w_t) - \mathrm{Id}, U_t\rangle \; . \tag{34}$$

Our main result in this section is the following, which says that it is possible to achieve such approximations with high probability in nearly linear time:

**Lemma 8.1.** *Let $S = \{X_1, \ldots, X_n\} \subseteq \mathbb{R}^d$, let $\delta > 0$, and let $w_0, \ldots, w_t$ be as above. Then there is an algorithm* APPROXSCORES$(S, w_0, \ldots, w_t, \delta)$ *which outputs $\tilde{\tau}_{t,i}$ and $\tilde{q}_t$ so that with probability $1 - \delta$, we have $\tilde{\tau}_{t,i} \approx_{0.1} \tau_{t,i}$ for all $i = 1, \ldots, n$, where $\tau_t, \tilde{q}_t$ are defined in (33) and (34), respectively. Moreover,* APPROXSCORES *runs in time $\widetilde{O}(tnd \log 1/\delta)$.*

If the output of APPROXSCORES satisfies the conditions of Lemma 8.1, we say that APPROXSCORES *succeeds*.

## 8.1 Tools from randomized numerical linear algebra

We need a few tools which are standard in the design of fast algorithms based on matrix multiplicative weights. The first is the standard Johnson-Lindenstrauss dimension reduction lemma:

**Lemma 8.2** (Johnson-Lindenstrauss lemma [27])**.** *Let $\Phi \in \mathbb{R}^{r \times d}$ be a matrix whose entries are i.i.d. samples from $\mathcal{N}(0, 1/r)$. For every vector $u \in \mathbb{R}^d$ and every $\epsilon \in (0, 1)$,*

$$\Pr\left[(1 - \epsilon)\|u\|_2 \leq \|\Phi u\|_2 \leq (1 + \epsilon)\|u\|_2\right] \geq 1 - \exp(-\Omega(\epsilon^2 r)) \,.$$

We also require the following, slightly stronger version of the JL guarantee, which states that it preserves matrix inner products:

**Lemma 8.3.** *Let $A, U \in \mathbb{R}^{d \times d}$. Suppose $U = BB^\top$ for some symmetric $B$. Let $S \in \mathbb{R}^{r \times d}$ have i.i.d. entries from $\mathcal{N}(0, 1/r)$. There is a universal constant $c$ such that for all $\epsilon > 0$,*

$$\Pr\left[|\langle A, BS^\top SB\rangle - \langle A, U\rangle| > \epsilon\|A\|_2 \cdot \operatorname{tr}(U)\right] \leq 2\exp(-cr \cdot \min(\epsilon, \epsilon^2)) \,.$$

*Proof.* Notice that $\mathbb{E}\left[\langle A, BS^\top SB\rangle = \langle A, U\rangle\right]$. By the Hanson-Wright inequality [28] together with standard arguments about averages of i.i.d. sub-exponential random variables, for every $t > 0$,

$$\Pr\left[|\langle A, BS^\top SB\rangle - \langle A, U\rangle| > t\right] \leq 2\exp(-\ell \cdot \Omega(\min(t^2/\|BAB\|_F^2, t/\|BAB\|_2))) \,.$$

To finish the proof it will be enough to show that

$$\|BAB\|_F^2 \leq \|A\|_2^2 \cdot \operatorname{tr}(U)^2 \quad \text{and} \quad \|BAB\|_2 \leq \|A\|_2 \cdot \operatorname{tr}(U) \,.$$

For the first statement, note that

$$\|BAB\|_F^2 = \operatorname{tr}(AUAU) = \langle AUA, U\rangle \leq \|AUA\|_2 \cdot \operatorname{tr}(U) \,.$$

Since spectral norm is sub-multiplicative, $\|AUA\|_2 \leq \|A\|_2^2\|U\|_2 \leq \|A\|_2^2 \cdot \operatorname{tr}(U)$.
    It remains to prove $\|BAB\|_2 \leq \|A\|_2 \cdot \operatorname{tr}(U)$. Again we have

$$\|BAB\|_2 \leq \|B\|_2^2\|A\|_2 = \|U\|_2\|A\|_2 \leq \operatorname{tr}(U) \cdot \|A\|_2 \,.$$

which finishes the proof. $\qquad\square$

We will also make use of Taylor series approximations to the matrix exponential function. The next lemma helps to control the errors incurred by such approximations.

**Lemma 8.4** (folklore, see e.g. [29])**.** *For $\ell \in \mathbb{N}$, let $P_\ell(Y) = \sum_{j=0}^{\ell} \frac{1}{j!}(Y)^j$ be the degree-$\ell$ Taylor series approximation to $\exp(Y)$. For every $n \times n$ symmetric real matrix $Y$,*

$$\|P_\ell(Y) - \exp(Y)\|_2 \leq \exp(\|Y\|_2) \cdot \exp(-\ell) \,.$$

## 8.2 Efficient approximate score oracles

With these tools in hand, we are now ready to describe APPROXSCORES. For some sufficiently large constant $C > 0$, let $\delta' = \delta/3$, let $r = C \log n/\delta'$, let $\ell = C \log d$, and let $S \in \mathbb{R}^{r \times d}$ be a matrix with i.i.d. entries from $\mathcal{N}(0, 1/r)$. The algorithm will form the $r \times d$ matrix

$$A_{r,\ell} = S \cdot P_\ell \left( -\frac{\alpha}{2} \sum_{t=0}^{t-1} M(w_t) \right) \ . \tag{35}$$

The estimate for the scores will then be given by

$$\tilde{\tau}_{t,i} = \frac{1}{\operatorname{tr}(A_{r,\ell} A_{r,\ell}^\top)} \left\| A_{r,\ell}(X_i - \mu(w_t)) \right\|_2^2 \ , \tag{36}$$

and the estimate for $q_{t,i}$ will be given by

$$\tilde{q}_{t,i} = \sum_{i=1}^{n} (\tilde{\tau}_{t,i} - 1) \ . \tag{37}$$

The formal pseudocode is given in Algorithm 5. We first demonstrate that APPROXSCORES indeed runs in

---

**Algorithm 5** Randomized nearly-linear time approximate score computation

1: **Input:** dataset $S \subset \mathbb{R}^d$ of size $n$, weight vectors $w_0, \ldots, w_t$, failure probability $\delta > 0$
2: Let $C > 0$ be a universal constant sufficiently large
3: Let $\delta' = \delta/3$, let $r = C \log n/\delta'$, and let $\ell = C \log d$.
4: Let $S \in \mathbb{R}^{r \times d}$ have entries drawn i.i.d. from $\mathcal{N}(0, 1/r)$.
5: Compute $A_{r,\ell}$ as defined as in (35).
6: For all $i = 1, \ldots, n$, let $\tilde{\tau}_{t,i}$ be as in (36), and let $\tilde{q}_t$ be as in (37).
7: **return** $\tilde{\tau}_t, \tilde{q}_t$

---

the claimed runtime:

**Lemma 8.5.** APPROXSCORES$(S, w_0, \ldots, w_t, \delta)$ *runs in time* $\widetilde{O}(tnd \log 1/\delta)$.

*Proof.* The main algorithmic work being done in APPROXSCORES is to form the matrix $A_{r,\ell}$. We claim that this matrix can be formed in time $\widetilde{O}(tnd \log 1/\delta)$. Afterwards, notice that $\operatorname{tr}(A_{r,\ell} A_{r,\ell}^\top)$ can be computed in time $O(dr^2)$, and given that, each $\tilde{\tau}_{t,i}$ can be computed in time $O(nr)$. Given the $\tilde{\tau}_{t,i}$, we then observe that $\tilde{q}_t$ can be computed in time $O(n)$. Since $r = O(\log n/\delta)$ and $\ell = O(\log d)$, we conclude that the overall runtime of APPROXSCORES is dominated by the time to form $A_{r,\ell}$, and so it runs in time $\widetilde{O}(tnd \log 1/\delta)$.

We now demonstrate how to form $A_{r,\ell}$ efficiently. Observe that for any vector $v \in \mathbb{R}^d$, we can evaluate $v^\top \sum_{i=0}^{t-1} M(w_i)$ in time $O(tnd)$. By iterating this process, we can compute $v^\top \left( \sum_{i=0}^{t-1} M(w_i) \right)^j$ in time $O(jtnd)$, and thus we can compute $v^\top P_\ell \left( -\frac{\alpha}{2} \sum_{i=0}^{t-1} M(w_i) \right)$ in time $O(\ell^2 tnd)$. Since $S$ has $r = O(\log n/\delta)$ rows, and $\ell = O(\log d)$, we can therefore form $A_{r,\ell}$ in time $\widetilde{O}(tnd \log 1/\delta)$ by forming each row of $A_{r,\ell}$. $\square$

Note that APPROXSCORES is an approximate augmented score oracle, and so it is also clearly an approximate score oracle. In the remainder of this section, we show:

**Lemma 8.6.** *With probability* $\geq 1 - \delta$*, the output of* APPROXSCORES *satisfies* $\tilde{\tau}_{t,i} \approx_{0.1} \tau_{t,i}$ *for all* $i = 1, \ldots, n$*, and* $\tilde{q}_t \approx_{0.1} q_t$.

*Proof.* Let $M = \sum_{i=0}^{t-1} M(w_t)$, and let $Y_t = \exp(M)$, so that $U_t = \frac{Y_t}{\mathrm{tr}(Y_t)}$, and so that $A_{r,\ell} = S \cdot P_\ell(M)$. Then, by Lemma 8.4 and our choice of $\ell$, we have $\left\| Y_t^{1/2} - P_\ell(M) \right\|_2 \leq \frac{0.01}{d} \cdot \left\| Y_t^{1/2} \right\|_2$, which immediately implies that $\left\| Y_t - P_\ell(M)^2 \right\|_2 \leq \frac{0.03}{d} \cdot \|Y_t\|_2$. Notice that this implies that $\mathrm{tr}(P_\ell(M)^2) \approx_{0.03} \mathrm{tr}(Y_t)$.

We now condition on the event that the following three events hold simultaneously:

$$\|SP_\ell(M)(X_i - \mu(w_t))\|_2^2 \approx_{0.01} \|P_\ell(M)(X - \mu(w_t))\|_2^2 \text{ for all } i = 1, \dots, n, \tag{38}$$

$$\mathrm{tr}(P_\ell(M)S^\top SP_\ell(M)) \approx_{0.01} \mathrm{tr}(P_\ell(M)^2), \tag{39}$$

$$\left| \langle M(w_t) - \mathrm{Id}, P_\ell(M)S^\top SP_\ell(M) \rangle - \langle M(w_t) - \mathrm{Id}, P_\ell(M)^2 \rangle \right| \leq 0.01 \cdot \mathrm{tr}(P_\ell(M)^2) \|M(w_t) - \mathrm{Id}\|_2. \tag{40}$$

By our choice of $\delta'$, Lemma 8.2, and a union bound, we know that (38) holds with probability at least $1 - \delta/3$. By instantiating Lemma 8.3 with $A = I$ and $A = M(w_t) - \mathrm{Id}$ respectively, we also know that (39) and (40) each hold with probability at least $1 - \delta/3$. Thus, by a union bound, all three conditions hold simultaneously with probability at least $1 - \delta$. We claim that conditioned on these three events, the conditions of the lemma are satisfied. Indeed, we have

$$\begin{aligned}
\tilde{\tau}_{t,i} &= \frac{1}{\mathrm{tr}(P_\ell(M)S^\top SP_\ell(M))} \|S \cdot P_\ell(M)(X_i - \mu(w_t))\|_2^2 \\
&\approx_{0.0121} \frac{1}{\mathrm{tr}(P_\ell(M)^2)} \|P_\ell(M)(X_i - \mu(w_t))\|_2^2 \\
&\approx_{0.0363} \frac{1}{\mathrm{tr}(Y_t)} \|P_\ell(M)(X_i - \mu(w_t))\|_2^2 \\
&\approx_{0.01} \frac{1}{\mathrm{tr}(Y_t)} (X_i - \mu(w)_t)^\top Y_t (X_i - \mu(w)_t) \\
&= \tau_{t,i}.
\end{aligned}$$

Here the second line follows from (38) and (39), the third line follows from our condition on the trace, and the final approximation follows since $P_\ell(M)$ approximates $Y_t$ in spectral norm. This proves the claim about the $\tilde{\tau}_{t,i}$.

To conclude, we observe that

$$\begin{aligned}
\tilde{q}_{t,i} &= \frac{1}{\mathrm{tr}(P_\ell(M)S^\top SP_\ell(M))} \sum_{i=1}^n \left( \|S \cdot P_\ell(M)(X_i - \mu(w_t))\|_2^2 - \mathrm{tr}(P_\ell(M)S^\top SP_\ell(M)) \right) \\
&= \frac{1}{\mathrm{tr}(P_\ell(M)S^\top SP_\ell(M))} \langle M(w_t) - \mathrm{Id}, P_\ell(M)S^\top SP_\ell(M) \rangle \\
&= \frac{1}{\mathrm{tr}(P_\ell(M)S^\top SP_\ell(M))} \left( \langle M(w_t) - \mathrm{Id}, P_\ell(M)^2 \rangle + \eta \right),
\end{aligned}$$

where $|\eta| \leq 0.01 \cdot \mathrm{tr}(P_\ell(M)) \cdot \|M(w_t) - \mathrm{Id}\|_2$ by (40). We further have

$$\begin{aligned}
\frac{1}{\mathrm{tr}(P_\ell(M)S^\top SP_\ell(M))} \langle M(w_t) - \mathrm{Id}, P_\ell(M)^2 \rangle &\approx_{0.01} \frac{1}{\mathrm{tr}(P_\ell(M)^2)} \langle M(w_t) - \mathrm{Id}, P_\ell(M)^2 \rangle \\
&\approx_{0.1} \langle M(w_t) - \mathrm{Id}, U_t \rangle \\
&= q_t.
\end{aligned}$$

Therefore

$$|\tilde{q}_t - q_t| \leq 0.1 q_t + \frac{|\eta|}{\mathrm{tr}(P_\ell(M)S^\top SP_\ell(M))} \leq 0.1 \cdot q_t + 0.05 \frac{|\eta|}{\mathrm{tr}(Y_t)} \leq 0.1 \cdot q_t + 0.05 \|M(w_t) - \mathrm{Id}\|_2,$$

which was the desired bound for $\tilde{q}_t$. $\qquad\square$

Lemmata 8.5 and 8.6 together immediately imply Lemma 8.1.

# 9 Putting it all together

In this section, we formally combine the guarantees derived in the previous sections to prove Theorems 4.1 and 4.2.

## 9.1 Proof of Theorem 4.1

Given the machinery we've developed, the algorithm is straightforward to describe. Given a corrupted dataset $S$ and $\delta > 0$, run $\textsc{NaivePrune}(S, \sqrt{4dn/\delta}, \delta/4)$ to obtain a pruned dataset $S'$. Center all points in $S'$ with the empirical mean of $S'$. Then, run $\textsc{QUEScoreFilter}(S', \textsc{ApproxScores})$, with $\kappa = \sqrt{4dn/\delta}$, and the $\delta$ parameter in $\textsc{ApproxScores}$ set to $O(\delta/(\log \kappa \log d))$. The formal pseudocode is presented in Algorithm 6.

---

**Algorithm 6** Nearly-linear time robust mean estimation under bounded second moments

---

1: **Input:** dataset $S \subset \mathbb{R}^d$ of size $n$, failure probability $\delta > 0$
2: Let $S' = \textsc{NaivePrune}(S, \sqrt{4dn/\delta}, \delta/4)$.
3: Let $\kappa = \sqrt{4dn/\delta}$.
4: Center all points in $S'$ at the empirical mean of $S'$.
5: Let $\mathscr{O}$ be $\textsc{ApproxScores}$ with failure probability $O(\delta/(\log \kappa \log d))$.
6: Let $\widehat{\mu} = \textsc{QUEScoreFilter}(S', \mathscr{O})$.
7: **return** $\widehat{\mu}$.

---

We now prove correctness.

*Proof of Theorem 4.1.* Recall that by definition, we may assume that $S = T \cup S_b \setminus S_r$, where $T$ is a set of $n$ i.i.d. samples from $D$, and $|S_b|, |S_r| \leq \varepsilon n$. Let $\gamma_1, \gamma_2$ be as in (9). We condition on four events:

- $\|X_i - \mu\|_2 \leq \sqrt{\frac{4dn}{\delta}}$ for all $i \in T$,

- $\textsc{NaivePrune}(S', \sqrt{4dn/\delta}, \delta/4)$ succeeds,

- $T = S_g \cup T_b$, where $S_g$ is $(\gamma_1, \gamma_2)$-good with respect to $D$, and $|T_b| \leq \varepsilon n$, and

- every time it is called, $\textsc{ApproxScores}$ suceeds.

By Chebyshev's inequality, and an union bound over all $n$ points in $T$, the first bullet point holds with probability at least $\delta/4$. By a further union bound and by adjusting constants in our choices of $\delta$, all four of these conditions hold simultaneously with probability at least $1 - \delta - \exp(-\varepsilon n)$. We now claim that, conditional on these four events, we output a $\mu(w)$ so that $\|\mu - \mu(w)\|_2 = O(\sqrt{\varepsilon}) + \widetilde{O}(\sqrt{d/(n\delta)})$. Indeed, the first two conditions imply that $\textsc{NaivePrune}$ does not throw away any points in $T$, and moreover, all points $X$ in the set $S'$ satisfy $\|X_i\|_2 \leq \sqrt{4dn/\delta}$ after centering. Thus, since the scores output by $\textsc{ApproxScores}$ satisfy the necessary conditions for Theorem 6.5, it follows that the final output satisfies $\|\mu - \mu(w)\|_2 = O(\sqrt{\varepsilon}) + \widetilde{O}(\sqrt{d/(n\delta)})$, as claimed.

We now turn to runtime. Since each epoch runs for at most $O(\log d)$ iterations, and so we run for at most $O(\log \kappa \log d)$ iterations, the total time spent running $\textsc{ApproxScores}$ is at most $\widetilde{O}(nd \log 1/\delta)$. Thus overall the algorithm runs in time $\widetilde{O}(nd \log 1/\delta)$, as claimed. □

## 9.2 Proof of Theorem 4.2

Again, the algorithm is straightforward. Given a corrupted dataset $S$, parameters $\varepsilon > 0$ and $\delta > 0$, run $\textsc{NaivePrune}(S, \sqrt{4d \log(n/\delta)}, \delta/4)$ to obtain a pruned dataset $S'$. Center all points in $S'$ with the empirical mean of $S'$. Then, as above, run $\textsc{s.g.-QUEScoreFilter}(S', \varepsilon, \textsc{ApproxScores})$, with $\kappa = \sqrt{4d \log(n/\delta)}$,

and the $\delta$ parameter in ApproxScores set to $O(\delta/(\log \kappa/\varepsilon \log d))$. The formal pseudocode is presented in Algorithm 7.

---

**Algorithm 7** Nearly-linear time robust mean estimation for isostropic subgaussian distributions

---

1: **Input:** dataset $S \subset \mathbb{R}^d$ of size $n$, failure probability $\delta > 0$, fraction of error $\varepsilon$
2: Let $S' = \text{NaivePrune}(S, \sqrt{4d\log(n/\delta)}, \delta/4)$.
3: Let $\kappa = \sqrt{4d\log(n/\delta)}$.
4: Center all points in $S'$ at the empirical mean of $S'$.
5: Let $\mathscr{O}$ be ApproxScores with failure probability $O(\delta/(\log \kappa/\varepsilon \log d))$.
6: Let $\widehat{\mu} = \text{s.g.-QUEScoreFilter}(S', \mathscr{O}, \varepsilon)$.
7: **return** $\widehat{\mu}$.

---

We now prove correctness. The proof is very similar to the proof presented above.

*Proof of Theorem 4.2.* Recall that by definition, we may assume that $S = S_g \cup S_b \setminus S_r$, where $T$ is a set of $n$ i.i.d. samples from $D$, and $|S_b|, |S_r| \leq \varepsilon n$. Let $\gamma_1, \gamma_2$ be as in (16). We condition on four events:

- $\|X_i - \mu\|_2 \leq \sqrt{4d\log(n/\delta)}$ for all $i \in S_g$,

- $\text{NaivePrune}(S', \sqrt{4d\log(n/\delta)}, \delta/4)$ succeeds,

- $S_g$ is $(\varepsilon, \gamma_1, \gamma_2)$-s.g. good with respect to $D$,

- every time it is called, ApproxScores suceeds.

By standard concentration inequalities for chi-squared random variables, and an union bound over all $n$ points in $T$, the first bullet point holds with probability at least $\delta/4$. By a further union bound and by adjusting constants in our choices of $\delta$, all four of these conditions hold simultaneously with probability at least $1 - \delta$. We now claim that, conditional on these four events, we output a $\mu(w)$ so that $\|\mu - \mu(w)\|_2 = O(\gamma_1 + \varepsilon\sqrt{\log 1/\varepsilon} + \sqrt{\varepsilon(\gamma_1 + \gamma_2)})$. The first two conditions imply that NaivePrune does not throw away any points in $S_g$, and moreover, all points $X$ in the set $S'$ satisfy $\|X_i\|_2 \leq \sqrt{4d\log(n/\delta)}$ after centering. By standard arguments, this implies that $\left\|M(\frac{1}{n}\mathbf{1}_n) - \text{Id}\right\|_2 \leq O(\varepsilon \log 1/\varepsilon + \varepsilon\kappa)$, for $\kappa = \sqrt{d\log(n/\delta)}$. Thus, since the scores output by ApproxScores satisfy the necessary conditions for Theorem 6.5, it follows that the final output satisfies $\|\mu - \mu(w)\|_2 = O(\gamma_1 + \varepsilon\sqrt{\log 1/\varepsilon} + \sqrt{\varepsilon(\gamma_1 + \gamma_2)})$, as claimed.

We now turn to runtime. Since each epoch runs for at most $O(\log d)$ iterations, and so we run for at most $O(\log \kappa/\varepsilon \log d)$ iterations, the total time spent running ApproxScores is at most $\widetilde{O}(nd \log 1/\delta \log 1/\varepsilon)$. Thus overall the algorithm runs in time $\widetilde{O}(nd \log 1/\delta \log 1/\varepsilon)$, as claimed. $\qquad\square$

# Part II: Outlier detection in high-dimensional data sets

## 10 QUE scoring versus local methods

The work [5] compares a number of outlier detection methods (mainly those based on $k$-NN distances) on several datasets, both low and high dimensional. We evaluate QUE scoring on the InternetAds dataset from [5], with a 0.1-fraction of outliers. Unlike the experiments on our CIFAR-10 and text embedding data, to replicate the experimental setting of prior work as closely as possible we perform no whitening or other preprocessing.

We find that QUE scoring is outperformed by LOF/$k$-NN-based methods on the InternetAds dataset. Choosing $\alpha = 4$ for QUE, we find the ROCAUC scores in the below table.

To elucidate the difference between the InternetAds setting where $k$-NN methods perform well and the other experimental settings in this paper, we offer the following histograms demonstrating that the

| method | ROCAUC |
| --- | --- |
| naive spectral | 0.539 |
| QUE | 0.626 |
| $\ell_2$ | 0.723 |
| isolation forest | 0.702 |
| local outlier factor | 0.853 |
| $k$-NN | 0.855 |

Figure 2: ROCAUC scores on InternetAds dataset from [5]. Note that QUE still improves on naive spectral methods, but because outliers are individually identifiable in the InternetAds data set, local and $\ell_2$ methods improve on any spectral method. We use $\alpha = 4$ in QUE. See [21, 22, 5] for definitions of local outlier factor and $k$-NN methods.

(a) InternetAds, $\epsilon = 0.1$      (b) synthetic, $\epsilon = 0.2$,

Figure 3: We plot histograms of the following collection $\{d_k(X_i, S) : X_i \in\}$ for $S = $ InternetAds,synthetic, where $d_k(X_i, S)$ is the average squared-$\ell_2$ distance of $X_i$ and its $k$ nearest neighbors, with $k = 10$. Observe that in the InternetAds dataset essentially only outliers have $d_k(X_i, S) > 15$, while for the synthetic dataset *all inliers $d_k(X_i, S)$ is much greater than that of every outlier.*

distribution of nearest-neighbor distances is markedly distinct for inliers and outliers in both data sets, but only in the InternetAds dataset do inliers have smaller $k$-NN distances.

# 11 Scaling up: a nearly-linear time implementation of QUE scoring

Most of the experiments we present involving QUE scores employ the following approach to compute them. Given $X_1, \ldots, X_n \in \mathbb{R}^d$, explicitly form the empirical covariance $\overline{\Sigma}$ in memory. Use SciPy's `expm` function to compute the matrix exponential $U = \exp(\alpha \overline{\Sigma})$, then compute $\tau_i = (X_i - \overline{\mu})^\top U (X_i - \overline{\mu})$. (This in turn uses the scaling and squaring algorithm for the matrix exponential of Al-Mohy and Higham [30].)

While we are already able to run experiments in 1000 or more dimensions using this approach, it requires at least $d^2$ memory to store the covariance, and somewhat more time to form and exponentiate it. We also implement an approximate method to compute QUE scores, whose running time is $\widetilde{O}(nd)$. We demonstrate in this section that outlier detection from approximate QUE scores still improves over baseline methods on several data sets. The technique here is very similar to the one used in Section 8 to approximate the scores used in the fast robust mean estimation algorithm. At a high level, the idea is the same: approximate the exponential with a low-degree polynomial, and sketch this using Johnson-Lindenstrauss matrices. However, we make a couple of additional optimizations here.

**QUE scores in nearly-linear time** We use the following approximate method, inspired by our sketching approach to compute QUE scores from our nearly linear time algorithm for robust mean estimation.

Given $X_1, \ldots, X_n$ and $\alpha > 0$, our goal is to compute approximations $\tilde{\tau}_i$ to the QUE scores. At a high level, our approach employs two main tricks:

1. Approximate the matrix exponential $\exp(M)$ by Chebyshev polynomials of degree $O(\log d)$.
2. For $\overline{\Sigma}$ the empirical covariance of $X_1, \ldots, X_n$, use fast versions of the Johnson-Lindenstrauss method to approximate $\langle X_i, \overline{\Sigma}^j X_i \rangle$ for all $i \le n$ and $j \le O(\log d)$.

Since for most applications only the *order* of the QUE scores matters, we will actually compute approximations to *non-normalized* scores, involving the matrix $U = \exp(\alpha \overline{\Sigma}/\|\overline{\Sigma}\|_2)$, rather than normalizing by $\text{tr}(U)$. For notational simplicity, let us assume $\overline{\mu} = 0$.

We first rewrite $\tau_i \propto \|\exp(\alpha \overline{\Sigma}/2\|\overline{\Sigma}\|_2) X_i\|_2^2$ where $\propto$ hides the normalization $\text{tr} \exp(\alpha \overline{\Sigma})$. If we approximate the matrix exponential by a degree $O(\log d)$ Chebyshev approximation $P$, the goal is now to approximate $\|P(\alpha \overline{\Sigma}/2\|\overline{\Sigma}\|_2) X_i\|^2$ for all $i$. Suppose $S$ is an $O(\log(d+n)) \times d$ sketching matrix. Then it will suffice to compute $M = SP(\alpha \overline{\Sigma}/2\|\overline{\Sigma}\|_2)$, since then in $\widetilde{O}(nd)$ time we can compute all the matrix-vector products $MX_i$ and their norms, and $\|SP(\alpha \overline{\Sigma}/2\|\overline{\Sigma}\|_2) X_i\|_2^2 \approx \|P(\alpha \overline{\Sigma}/2\|\overline{\Sigma}\|_2) X_i\|_2^2$.

Note that using the expansion into powers $\overline{\Sigma}^j$, right or left matrix-vector multiplication by $P(\alpha \overline{\Sigma}/2\|\overline{\Sigma}\|_2)$ can be accomplished in $O(nd \log d)$ time. As in the fast JL transform [31], we take $S = S' \cdot D \cdot H$ where $S'$ is a sparse random matrix, $D$ is a diagonal matrix with random $\pm 1$ entries, and $H$ is a Hadamard matrix. Fast fourier transform methods may be used to compute matrix-vector multiplications $SX$ in $d(\log d)^{O(1)}$ time [32], leading to nearly-linear running time of this approach in theory. In practice, we use standard matrix multiplication; this still allows for experiments in thousands of dimensions.

Experiments show that the approximate QUE scores are consistent with the exact QUE scores, in that higher $k$ leads to larger maximum improvement over naive spectral scores; furthermore approximate QUE consistently interpolates between $\ell_2$ scores (when $\alpha = 0$) and naive spectral scores (when $\alpha = \infty$).

(a) synthetic 128-dimensions      (b) synthetic 128-dimensions

Figure 4: We plot the improvement of ROCAUC scores of our approximate QUE scoring implementation, over baseline methods $\ell_2$ scoring as well as naive spectral scoring, on **128-dimensional** synthetic data, for 3, 6, and 10 directions of corruption. We use the degree-5 Chebyshev approximation to the matrix exponential function, along with scaling and squaring for approximating outside the interval $[-1, 1]$.

(a) synthetic 8192-dimensions      (b) synthetic 8192-dimensions

Figure 5: Similar to above, these show the improvement of ROCAUC scores of our approximate QUE scoring implementation, over baseline methods $\ell_2$ scoring and naive spectral scoring, on **8192-dimensional** synthetic data, for 12 and 15 directions of corruption. These approximate QUE scores run in nearly-linear time and improve over the performance of naive spectral and $\ell_2$-based scoring.

## Footnotes

[1]Some prior algorithms, e.g. the *filter* of [1] instead iteratively throw out points suspected to be outliers. However, since those algorithms are (necessarily) randomized, they can also be viewed as weighting points, where the weight of $X_i$ is the probability it has not been thrown out. The algorithm we present here can also be implemented by throwing out points in a randomized fashion – we discuss further in supplementary material.

[2]This is true in the real RAM model; in practice we can run any iteration in $O(m \log(\tau_{\max}/\sigma))$ time by exponentiation via repeated doubling to compute all the $w_i^{(t)}$, so we pay at most an additional log factor

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

# A    Deferred details from Section 4

## A.1    Proof of Lemma 5.1

The algorithm is straightforward: choose a random point in $S$, and check if strictly more than $n/2$ points lie within a ball of radius $2r$ around this point. If so, include all points with distance at most $4r$ from this point. If not, repeat, and run for $O(\log 1/\delta)$ iterations. We now prove correctness.

*Proof of Lemma 5.1.* By the triangle inequality, if we ever randomly select a point from $S'$, then we terminate, and in this case it is easy to see that the output satisfies the desired property. Thus, it is easy to see that the probability we have not terminated after $t$ iterations is at most $2^{-t}$. Suppose we have terminated. Then in that iteration, we selected a point $X \in S$ that has distance at most $2r$ to more than $n/2$ other points in $S$. This implies that it has distance at most $2r$ to some point in $S'$. By triangle inequality, this implies that all points in $S'$ are at distance at most $4r$ from $X$, and so the output in this iteration must satisfy the claims of the Lemma.  □

We note that if one wishes to obtain a deterministic linear-time algorithm for this problem, it is also possible to do so, albeit using radius $O(r\sqrt{d})$. The algorithm is again simple: simply take the coordinate-wise median of all the data points, and take all points with distance at most $O(r\sqrt{d})$ from this point. It is not hard to see that the coordinate-wise median can differ in each coordinate from the points in $S'$ by at most $r$, and so its distance to each point in $S'$ can be at most $r\sqrt{d}$. While this is worse by a polynomial factor than the guarantee obtained above, since in the end our overall guarantees depend only logarithmically on $r$, this does not change our runtime guarantees by more than a logarithmic factor.

## A.2 Omitted details from Section 5.3

The algorithm 1DFILTER is quite simple. For $i = 1, \ldots, m$, and for any positive integer $t$, define

$$w_i^{(t)} = \left(1 - \frac{\tau_i}{\tau_{\max}}\right)^t w_i \text{ , and } F_t = \sum_{i=1}^n w_i^{(t)} \tau_i \ . \tag{41}$$

Observe that the $F_t$ form a monotone decreasing sequence. The algorithm will simply find the smallest $t \in \{1, \ldots, \frac{\tau_{\max}}{eb\sigma}\}$ so that $F_t \leq b\sigma$ via binary search, and outputs $w^{(t)}$. The formal pseudocode for the algorithm is given in Algorithm 8.

---

**Algorithm 8** Improved univariate score downweighting

1: **Input:** nonnegative scores $\tau_1, \ldots, \tau_m$, weights $w_1, \ldots, w_m$, parameters $b, \eta$
2: Let $\tau_{\max} = \max_{i \in [m]} \tau_i$.
3: Let $\sigma = \sum_{i=1}^m w_i \tau_i$.
4: By binary search, find the smallest $t \in \{1, \ldots, \frac{\tau_{\max}}{eb\sigma}\}$ satisfying $F_t \leq b\sigma$, where $F_t$ is defined in (41)
5: **return** The weights $w_1^{(t)}, \ldots, w_m^{(t)}$, where $w_i^{(t)}$ is defined as in (41).

---

We now prove that this algorithm satisfies Theorem 5.2. We say any set of weights $w'$ satisfying $\sum_{i \in S_g} w_i - w_i' \leq \sum_{i \in S_b} w_i - w_i'$ is *admissible*. We first show that the sequence of weights we produce is always admissible, under some mild conditions:

**Lemma A.1.** *Let $t$ be an integer so that $w^{(t)}$ is admissible, and $F_t > 2\eta\sigma$. Then $w^{(t+1)}$ is admissible.*

*Proof of Lemma A.1.* Because $w^{(t)} \leq w$, we have that $\sum_{i \in S_g} w_i^{(t)} \tau_i \leq \eta\sigma$. As a result, if $F_t \geq 2\eta\sigma$, we must have $\sum_{i \in S_b} w_i^{(t)} \tau_i > \sigma/2$. Therefore, we have the following two inequalities:

$$\sum_{i \in S_g} w_i^{(t)} - w_i^{(t+1)} = \frac{1}{\tau_{\max}} \sum_{i \in S_g} w^{(t)} \tau_i \leq \frac{\sigma}{2\tau_{\max}}$$

$$\sum_{i \in S_b} w_i^{(t)} - w_i^{(t+1)} = \frac{1}{\tau_{\max}} \sum_{i \in S_b} w^{(t)} \tau_i > \frac{\sigma}{2\tau_{\max}} \ .$$

Consequently, we remove more mass from the weights in $S_b$ than from $S_g$ in going from $w^{(t)}$ to $w^{(t+1)}$. Since by assumption $w^{(t)}$ is admissible, this immediately implies that $w^{(t+1)}$ is admissible as well. $\square$

We are now ready to prove Theorem 5.2.

*Proof of Theorem 5.2.* By induction, Lemma A.1 guarantees that the output weights remain admissible, and the termination condition of the algorithm guarantees that the output satisfies (5). It suffices to bound the runtime of the algorithm.

First, observe that there must exist a valid $T$ in the range we are searching. We first observe that

$$w_i^{(t)} \tau_i = w_i \left(1 - \frac{\tau_i}{\tau_{\max}}\right)^t \tau_i$$

$$\leq w_i \exp\left(-\frac{t \cdot \tau_i}{\tau_{\max}}\right) \tau_i \ .$$

For any constant $A > 0$, the maximizer of the function $g(x) = x \exp(-Ax)$ in the range $x \in [0, \infty)$ is achieved by $x = \frac{1}{A}$, so $g(x) \leq \frac{1}{eA}$ for all $x \in [0, \infty)$. Setting $A = t/\tau_{\max}$, and letting $t = \frac{\tau_{\max}}{eb\sigma}$, we conclude that

$$\sum_{i=1}^m w_i^{(t)} \tau_i \ \leq \sum_{i=1}^m w_i \cdot \frac{\tau_{\max}}{et} \leq b\sigma \ .$$

Thus, there exists some $t$ within our specified range which satisfies the conclusion. Finally, to bound the runtime, observe that every iteration runs in $O(n)$ time,[2] and we can run for at most $O(\log \tau_{\max}/(b\sigma))$ iterations, which completes the proof. $\qquad\qquad\qquad\qquad\qquad\qquad\qquad\qquad\qquad\qquad\qquad\qquad\qquad\quad\ \Box$

## A.3 The randomized hard filter

In this section we show that a randomized outlier removal method, rather than soft downweighting, can achieve the more or less the same guarantees as 1DFILTER. Formally, we show:

**Theorem A.2.** *Let $\eta \in (0, 1/2)$, let $b \geq 2\eta$, and let $s, \delta > 0$. Let $m$ satisfy*

$$ m = \widetilde{\Omega}\left( \frac{s \log^2(1/\delta) \log \frac{\tau_{\max}}{b\sigma}}{\varepsilon} \right) . $$

*Let $\tau_1, \ldots, \tau_m$ be non-negative scalars, and let $\tau_{\max} = \max_{i \in [m]} \tau_i$. Suppose there exist two disjoint sets $S_g, S_b$ so that $S_g \cup S_b = [m]$, and moreover,*

$$ \sum_{i \in S_g} \tau_i \leq \eta\sigma \ , \ where \ \ \sigma = \sum_{i=1}^{n} \tau_i \ . $$

*Then* RANDOMFILTER$(w, \tau)$ *runs in time $O\left(\left(1 + \log \frac{\tau_{\max}}{b\sigma}\right) m\right)$ and outputs $S' \subseteq S$ so that with probability $1 - \delta$, we have:*

- *not too many more points from $S_g$ are removed than from $S_b$ i.e.*

$$ |(S \setminus S') \cap S_g| \leq |(S \setminus S') \cap S_b| + \frac{\varepsilon m}{s} \ \ , \ and $$

- *the sum of the $\tau$ has decreased, i.e. $S'$ satisfies*

$$ \sum_{i \in S'} \tau_i \leq b\sigma \ . \tag{42} $$

The algorithm itself is very easy to describe. First, let $T = [m]$. Then, while $\sum_{i \in T} \tau_i > b\sigma$, throw away each point from $T$ with probability $\tau_i/\tau_{\max}(T)$, where $\tau_{\max}(T) = \max_{i \in T} \tau_i$, and let $T$ be the set of remaining points. At termination, we simply output the set $S' = T$. The formal pseudocode of this algorithm is given in Algorithm 9.

---
**Algorithm 9** Improved randomized univariate filtering
---
1: **Input:** nonnegative scores $\tau_1, \ldots, \tau_m$, parameters $b, \eta$
2: Let $T = [m]$
3: Let $\sigma = \sum_{i=1}^{n} \tau_i$.
4: **while** $\sum_{i \in T} \tau_i \geq b\sigma$ **do**
5: $\quad$ Let $\tau_{\max} = \max_{i \in T} \tau_i$
6: $\quad$ Remove each point $i \in T$ from $T$ with probability $\tau_i/\tau_{\max}$
7: **end while**
8: **return** The set $T$
---

As a brief aside, we note that this differs slightly from the algorithm presented in [4], as there the algorithm randomly selects a threshold, and throws away all points above this threshold. However, the key

property which the previous algorithm used of this random threshold was that for all $i \in T$, we had that $\Pr[i \text{ is thrown out}] = \tau_i/\tau_{\max}(T)$. Therefore a very similar analysis can be adapted for either case. However, the algorithm in [4] only succeeds with constant probability, and a martingale-style argument (as in [4]) is needed to ensure that it works.

The remainder of this section is dedicated to the proof of Theorem A.2. Our first lemma is similar to Lemma A.1.

**Lemma A.3.** *Suppose that $\sum_{i \in T} \tau_i > b\sigma$. Then, if we let $T'$ be the random set obtained by throwing away each point from $T$ with probability $\tau_i/\tau_{\max}(T)$, then:*

$$\mathbb{E}\left[|(T \setminus T') \cap S_g|\right] < \mathbb{E}\left[|(T \setminus T') \cap S_b|\right] , \tag{43}$$

*and moreover, for any $s > 0$, we have*

$$\Pr\left[|(T \setminus T') \cap S_g| > |(T \setminus T') \cap S_b| \text{ and } |T \setminus T'| > \frac{\varepsilon m}{s}\right] \le \exp\left(-\Omega\left(\frac{\varepsilon m}{s} \cdot \min\left((b - 2\eta)^2, 1\right)\right)\right) . \tag{44}$$

*Proof.* For $i \in T$, let $Y_i$ be the random variable which is 1 if we throw out $i$ in $T'$ and 0 otherwise. Then

$$\mathbb{E}\left[|(T - T') \cap S_g|\right] = \mathbb{E}\left[\sum_{i \in S_g \cap T} Y_i\right] = \sum_{i \in S_g \cap T} \frac{\tau_i}{\tau_{\max}(T)} \le \sum_{i \in S_g} \frac{\tau_i}{\tau_{\max}(T)} \le \frac{1}{\tau_{\max}(T)}\eta\sigma ,$$

$$\mathbb{E}\left[|(T - T') \cap S_b|\right] = \mathbb{E}\left[\sum_{i \in S_b \cap T} Y_i\right] = \sum_{i \in S_b \cap T} \frac{\tau_i}{\tau_{\max}(T)} \ge \frac{1}{\tau_{\max}(T)}(\beta - \eta)\sigma ,$$

which proves (43), since $\beta > 2\eta$.

To prove (44), we break into two cases depending on $\sigma/\tau_{\max}$. Suppose that $\sigma/\tau_{\max} \ge \varepsilon m/s$. Then by Bernstein's inequality, we have

$$\Pr\left[|(T \setminus T') \cap S_g| > |(T \setminus T') \cap S_b| \text{ and } |T \setminus T'| > \frac{2\varepsilon m}{s}\right] \le \Pr\left[|(T \setminus T') \cap S_g| > |(T \setminus T') \cap S_b|\right]$$

$$= \Pr\left[\sum_{i \in S_g \cap T} Y_i - \sum_{i \in S_b \cap T} Y_i > 0\right]$$

$$\le \exp\left(-\Omega\left(\frac{\sigma}{\tau_{\max}} \cdot \min\left((b - 2\eta)^2, b - 2\eta\right)\right)\right)$$

$$\le \exp\left(-\Omega\left(\frac{\varepsilon m}{s} \cdot \min\left((b - 2\eta)^2, b - 2\eta\right)\right)\right)$$

Otherwise, we observe that

$$\Pr\left[|(T \setminus T') \cap S_g| > |(T \setminus T') \cap S_b| \text{ and } |T \setminus T'| > \frac{2\varepsilon m}{s}\right] \le \Pr\left[|T \setminus T'| > \frac{\varepsilon m}{s}\right]$$

$$= \Pr\left[\sum_{i \in T} Y_i > \frac{2\varepsilon m}{s}\right]$$

$$\le \exp\left(-\Omega\left(\frac{\varepsilon m}{s}\right)\right) .$$

Combining these two cases, and simplifying yields the desired claim. $\qquad\square$

We now show that with high probability, we do not need to repeat this procedure too many times before the sum decreases by a constant factor.

**Lemma A.4.** *Let $\delta > 0$, and let $t = \widetilde{\Omega}\left(\log\left(\frac{\tau_{\max} m}{b\sigma}\right)\log(1/\delta)\right)$. Then, the probability that Algorithm 9 runs for more than $t$ iterations is at most $\delta$.*

*Proof.* Let $J = \log\frac{\tau_{\max} m}{b\sigma}$. For $j = 1, \ldots, J$, let

$$A_j = \left\{i \in T : \tau_i \in \left(\tau_{\max} \cdot 2^{-j}, \tau_{\max} \cdot 2^{-j-1}\right]\right\} .$$

For all $j = 1, \ldots, J$, we claim that conditional on the event that the algorithm has not terminated yet, and all points from $A_{j'}$ have been removed, for $j' < j$, then after $t'$ iterations, all points from $A_j$ have been removed with probability at least $1 - m2^{-t'}$. Indeed, for all $i \in A_j$, in every iteration, if it has not been already removed, then it is removed with probability at least $1/2$. Thus after $t'$ iterations, the probability that any point from $A_j$ remains is at most $n2^{-t'}$. Therefore, by a union bound, after $Jt'$ iterations, conditioned on the event that the algorithm hasn't terminated yet, the probability that any point from $A_j$ for any $j$ is at most $Jm2^{-t'}$. However, if all points from $A_j$ are removed, for all $j$, then if $T'$ is the remaining set, we have $\sum_{i \in T'} \tau_i \leq b\sigma$, so if all such points are removed, then the algorithm must either terminate or have already terminated. This proves the claim by setting $t' = \log(Jm/\delta)$. $\qquad\square$

Theorem A.2 follows from Lemma A.3 and Lemma A.4, by appropriately adjusting parameters.

**The full algorithm using the randomized filter** It is straightforward for the full matrix multiplicative weights algorithm to use the randomized filter rather than the downweighting-based method. Given an $\varepsilon$-corrupted dataset of size $n$ initally, we do the following. At every instance where we pass to the filter, simply run the randomized filter instead of the downweighting-based method, and output the set of weights which is $1/n$ for every point that remains after running the randomized filter, and 0 otherwise.

The guarantee of the randomized filter is slightly weaker than the guarantee of the downweighting-based method, so we cannot use black-box use the analysis presented beforehand to also analyze the algorithm instantiated with the weights given by the randomized filter. This is because our guarantee allows for slightly more good points than bad points to be removed per run of the algorithm. However, since the matrix multiplicative weights routine runs for at most polylogarthmically many iterations, by setting $s = \text{poly}\log(nd)$, we can guarantee that at the end of all of the runs, we have removed at most $2\varepsilon n$ data points from $S_g$. It is straightforward to verify that (up to a factor of 2), the same analysis for matrix multiplicative works with this slightly weaker guarantee, for $\varepsilon$ sufficiently small. For conciseness, we omit the proof.

# B  Omitted Proofs from Section 6

## B.1  Proof of Lemma 6.1

Before we prove this lemma, we require the following matrix Chernoff bound:

**Fact B.1** (Theorem 5.1.1 in [33]). *Let $M_1, \ldots, M_n \in \mathbb{R}^{d \times d}$ be a sequence of independent, random, PSD matrices. Assume that $\|M_i\|_2 \leq L$ for all $i = 1, \ldots, n$, and suppose $\|\mathbb{E}\left[\sum_{i=1}^n M_i\right]\|_2 \leq n$. Then, there is some universal constant $c \leq 2\log 2 - 1$ so that for all $t \geq 2$, we have*

$$\Pr\left[\left\|\sum_{i=1}^n M_i\right\|_2 \geq tn\right] \leq d\exp\left(-ctn/L\right) .$$

*Proof.* Observe that if $X \sim D$, then $\mathbb{E}\left[\|X - \mu\|_2^2\right] = \text{tr}(\Sigma) \leq d$. Let $c > 0$ be a constant to specify later. By Markov's inequality, we have

$$\Pr\left[\|X - \mu\|_2 \geq \sqrt{\frac{d}{c\varepsilon}}\right] \leq c\varepsilon . \tag{45}$$

Let $E$ be the event $E = \left\{ X : \|X - \mu\|_2 < \sqrt{\frac{d}{c\varepsilon}} \right\}$, and let $S = \{X_i : X_i \in E\}$. We claim this set satisfies the properties claimed.

We first demonstrate that with probability $1 - \exp(-\varepsilon n)$, we have $|S| \geq (1-\varepsilon)n$. Letting $Z_i = \mathbb{I}\{X_i \in E^c\}$, we have we have $|S| = n - \sum_{i=1}^n Z_i$, where $\mathbb{E}[Z_i] \leq c\varepsilon$, and hence, by Chernoff bounds,

$$\Pr\left[|S| < (1-\varepsilon)n\right] = \Pr\left[\sum Z_i > \varepsilon n\right]$$
$$\leq \left(\frac{e^{1/c-1}}{(1/c)^{1/c}}\right)^{c\varepsilon n}$$
$$= \exp\left(\varepsilon n - c\varepsilon n - \varepsilon n \log 1/c\right) .$$

In particular, we let $c = e^{-2}$, by simplifying we obtain that $\Pr\left[|S| < (1-\varepsilon)n\right] < \exp(-\varepsilon n)$. Let $E_1$ be the event that $|S| \geq (1-\varepsilon)n$.

We now demonstrate that with probability $1 - \delta$, our set $S$ is $(\gamma_1, \gamma_2)$-good with respect to $D$, where $\gamma_1, \gamma_2$ are given as in (9). We first prove concentration of the empirical mean. For $i = 1, \ldots, n$, let $Y_i = (X_i - \mu) \cdot \mathbb{I}\{X_i \in E\}$. Since the $X_i$ are independent, so too are the $Y_i$. Moreover, since multiplying by an indicator variable can only decrease variance, we have that $\text{Cov}[Y_i] \preceq I$. Letting $\mu' = \mathbb{E}[Y_i]$, we thus have

$$\mathbb{E}\left[\left\|\frac{1}{n}\sum_{i=1}^n Y_i - \mu'\right\|_2^2\right] \leq \frac{\text{tr}(\Sigma)}{n} \leq \frac{d}{n} .$$

Thus, if we let $E_2$ be the event

$$E_1 = \left\{\left\|\frac{1}{n}\sum_{i=1}^n Y_i - \mu'\right\|_2 \leq \sqrt{\frac{2d}{n\delta}}\right\} ,$$

by Markov's inequality, we have $\Pr[E_1] \geq 1 - \delta/2$. We additionally have that for any unit vector $v$,

$$|\langle v, \mu'\rangle| = |\mathbb{E}\left[\langle v, X_i - \mu\rangle \cdot \mathbb{I}\{X_i \in E\}\right]|$$
$$= |\mathbb{E}\left[\langle v, X_i - \mu\rangle \cdot \mathbb{I}\{X_i \in E^c\}\right]|$$
$$\leq \mathbb{E}\left[\langle v, X_i - \mu\rangle^2\right]^{1/2} \Pr_{X\sim D}[X \in E^c]^{1/2}$$
$$= \left(v^T \Sigma v\right)^{1/2} \cdot \Pr_{X\sim D}[X \in E^c]^{1/2} \leq \sqrt{c\varepsilon} ,$$

where the third line follows from Cauchy-Schwartz, the last line follows from (45), and since $\Sigma \preceq \text{Id}$. Taking a supremum over all unit vectors $v$ yields that $\|\mu'\|_2 \leq \sqrt{c\varepsilon}$. Therefore, conditioned on both $E_1$ and $E_2$, we have

$$\left\|\frac{1}{|S|}\sum_{i\in S} X_i - \mu\right\|_2 = \frac{n}{|S|}\left\|\frac{1}{n}\sum_{i=1}^n Y_i\right\|_2$$
$$\leq \frac{n}{|S|}\left(\left\|\frac{1}{n}\sum_{i=1}^n Y_i - \mu'\right\|_2 + \|\mu'\|_2\right)$$
$$\leq \frac{n}{|S|} \cdot \left(\sqrt{\frac{2d}{n\delta}} + \sqrt{c\varepsilon}\right)$$
$$\leq \frac{1}{1-\varepsilon} \cdot \left(\sqrt{\frac{2d}{n\delta}} + \sqrt{c\varepsilon}\right) = \gamma_1 . \qquad (46)$$

We now turn our attention to the claimed bound on the covariance. Let $Y_i$ be as above. Then, by assumption we have $\left\|Y_i Y_i^\top\right\|_2 = \left\|Y_i\right\|_2^2 \leq \frac{d}{c\varepsilon}$, and moreover we have

$$\mathbb{E}\left[\sum_{i=1}^n Y_i Y_i^\top\right] \preceq n \mathrm{Cov}\left[Y_i\right] \leq n \cdot \mathrm{Id} \ .$$

Hence, by Fact B.1, we have that

$$\Pr\left[\left\|\frac{1}{n}\sum_{i=1}^n Y_i Y_i^\top\right\|_2 > \frac{d(\log d + \log 2/\delta)}{c'\varepsilon n}\right] \leq \delta/2 \ , \tag{47}$$

for some universal constant $c' > 0$. Let $E_3$ be the event that (47) holds. Then, conditioned on both $E_1$ and $E_3$ holding, we have

$$\begin{aligned}
\|\mathrm{Cov}(S)\|_2 &= \left\|\frac{1}{|S|}\sum_{i=1}^n (Y_i - \mu(S))(Y_i - \mu_S)^\top\right\|_2 \\
&\overset{(a)}{\leq} \left\|\frac{1}{|S|}\sum_{i=1}^n Y_i Y_i^\top\right\|_2 \\
&\leq \frac{1}{1-\varepsilon} \cdot \frac{d(\log d + \log 2/\delta)}{c'\varepsilon n} = \gamma_2 \ ,
\end{aligned} \tag{48}$$

as claimed, where (a) follows since centering the second moment matrix can only decrease its top eigenvalue. Thus, (46) and (48) imply that, conditioned on events $E_1, E_2, E_3$ simultaneously, the set $S$ is $(\gamma_1, \gamma_2)$-good with respect to $D$. By a union bound, these three events happen simultaneously with probability at least $1 - \delta - \exp(-\varepsilon n)$, as claimed. $\qquad\square$