[Reviews · NeurIPS 2019]

Reviewer 1



The paper studies primarily studies the problem of robust mean-estimation, when a certain fraction of the data is corrupted adversarially. Then, in this setting, the authors propose a nearly-linear time estimator(which is also practical), which achieves the information-theoretic optimal rate. The authors back their claims by conducting experiments on CIFAR, word embeddings etc. These are some of the first experiments to be done at this scale in the robust statistics community. The paper is well written and the authors provide a lot of intuition for their algorithms. The authors are very rigorous in their theoretical claims for robust mean estimation. A minor criticism is the handling of the "outlier detection component". I agree with the authors that prima-facie it is not clear, when the outlier detection problem is meaningful. But, then, what is the the output of the algorithm 2 giving? From my understanding, I can see that if the outlier distribution is such that operator norm of the covariance of the mixture increases, then the weighting given by algorithm 2, can be used to identify the outlier points -- is this correct? But is this an if and only if? For example, say the outlier distribution N, is a point-mass at the true mean of D, then what would algorithm 2 give? Minor Question: --In step 3 of algorithm 2, is there a centering step missing?

Reviewer 2



This work considers the problem of identifying adversarial outliers, with applications to robust mean estimation. The work proposes to use a "quantum entropy" regularization which can be solved quickly using a matrix multiplicative weights algorithm. This leads to a near-linear time algorithm. The paper has a nice high-level discussion of how the entropy term improves upon the previous, spectral filtering methods and obtains the improved running time: intuitively, whereas the spectral filtering corrects a single direction at a time, the entropy term encourages filtering many directions simultaneously. Obtaining a (near-)linear time algorithm is pretty significant, as the time complexity significantly impacts the breadth of settings in which such methods may be applied (and of course, we can't hope to beat linear time). Large data sets demand linear time algorithms. The paper also includes experiments demonstrating that the method is effective. The experiments do convincingly show that the method is effective at removing outliers compared to several simple methods. The experiments are pretty well constructed, considering some natural models of contamination in two real data sets (CIFAR and word embeddings) and a synthetic data set. Compared to the simple (fast) baseline methods considered here, the new method does show significant improvement at identifying outliers. I have a few small criticisms here, though. First, even though the polynomial-time algorithms proposed in prior works are not near-linear time, is it infeasible to run them on these data sets? Since the running time guarantee was the main theoretical focus of the work, I would have liked to see some evaluation of the running times in addition to the accuracy; conversely, it would have been interesting to see the AUC scores achieved by these slower polynomial-time methods. If they are the same, then the new method should be a clear "win," but even if they achieved a better AUC in practice, it is desirable to understand what the trade-off is. I realize that the baseline methods have been chosen because they are fast, but they seem a bit weak, and it would be good to include a strong-but-slow baseline to understand if the new method gives anything up in practice. Finally, the plots for the experiments are so small that they are unreadable unless one zooms in or uses a magnifying glass. There is no point to including them in a print version. As nice as the extensive plots are, it would be better to choose a (representative) selection and just note that the others are similar.

Reviewer 3



Originality: The work proposes a new technique based on outlier scoring for robust estimation and it intuitively builds on the ideas of previous works. This can also be used for outlier detection. It also cites the related works adequately. Because of being the first (nearly) linear time algorithm, I think it scores well on originality. Quality: Experiments: I think there should be more experiments for robust mean estimation (see the improvements section for comments on this). Proofs: The authors have provided proofs for their claims. I found a few typos (see the improvement section for comments on this). I also have one question for the authors - for inequality (b) in (21) how did you bound \|\rho\|_2 \geq 1? Clarity: The paper is well written and well organized. However, because there are so many symbols and definitions, I would recommend a table pointing to definitions of terms like Goodness of Set,M, U. Also, it would be nice to explicitly define S_r and S_b ( I couldn't find them explicitly defined somewhere.) Significance: The significance is high because this paper gives the first linear time algorithm for robust mean estimation and the technique used seems intuitive. Because this is linear in time and thus practical, it may benefit the robust machine learning research in a variety of ways.

[Author Response · NeurIPS 2019]

We thank all the reviewers for their thorough and helpful remarks. We begin by addressing two main points, regarding theory for outlier detection (Reviewer 1) and additional experiments (Reviewers 2,3).

**Theory for Outlier Detection**    We agree with Reviewer 1 that the outlier detection problems solved by our algorithm are exactly those where the outliers increase the operator norm of the empirical covariance from what it would have been with a clean version of the data set. (One might call them "spectral outliers".) These outlier detection problems are motivated in part by hard instances of the robust mean estimation problem – those on which algorithms preceding the 2016 works of Lai-Rao-Vempala and Diakonikolas-Kamath-Kane-Li-Moitra-Stewart would have incurred large errors. The example suggested by Reviewer 1, where the outlier distribution is a point-mass at the true mean of the inlier distribution, does not fit this spectral outliers setting. However, in this example, many statistics can be accurately estimated without removing the outliers (e.g. the empirical mean and covariance will be good estimators of the population mean and covariance, respectively).

It is possible to formally state and prove such a result characterizing spectral outliers and our algorithm's success at such outlier detection tasks. Simple generative models (such as those we use to create synthetic data sets used in our experiments) lead to data sets with spectral outliers where our algorithm provably finds more outliers than e.g. naive spectral methods, outlier detection based on $\ell_2$ norms, etc. (The proofs would use standard spectral analysis and concentration of measure – a small subset of the techniques used to prove our results on robust mean estimation.)

We elected not to formally define the task of outlier detection – we believe that any mathematical definition will fail to capture the breadth of real-world situations where outlier detection might be used. (By contrast, in the more theoretical portion of the paper on robust mean estimation, we have formal problem definitions and rigorous theorem statements.) That said, in any final version of our paper we will clarify with increased mathematical formality the subset of outlier detection tasks for which our outlier detection algorithm succeeds.

**Robust Mean Estimation Experiments** Reviewers 2 and 3 suggest additional experiments (1) comparing our algorithms against slower baselines for outlier detection, (2) measuring running time, and (3) on robust mean estimation in addition to outlier detection. We first note that we did compare against state-of-the-art methods based on *local outlier factors* – many of these algorithms do not run in linear time. See plots (g),(h),(i) and section 10 of supplementary material. The *naive spectral* algorithm which we compared against (captured by $\alpha \to \infty$ in our plots) is the natural outlier detection analogue of the slow-but-polynomial-time algorithms for robust mean estimation preceding our work – we observed experimentally that QUE scoring has improved AUC scores compared to this algorithm.

In preparing this author response we conducted some preliminary experiments on *robust mean estimation* using our synthetic data set. We compared two iterated filtering methods, one based on QUE scores and one based on naive spectral scores (the case of $\alpha \to \infty$) as used in prior work on polynomial-time algorithms for robust mean estimation (Figure ). We iterated the filter, computing scores, removing a small fraction of the data points with the highest scores, and iterating until the empirical covariance had bounded spectral norm. We compared the accuracy (in Euclidean norm) and running times of both methods – both were implemented with optimized matrix multiplication libraries (BLAS) under PyTorch; we used our fast approximate QUE implementation (Section 11, supplement). To ensure a fair comparison, we did a hyperparameter sweep to optimize the runtime and accuracy of naive spectral filtering, and only included a single setting of hyperparameters for QUE scoring.

Our findings confirm the theory in our paper: although each iteration of QUE scoring is slightly slower than naive spectral scoring, the latter requires many more iterations to find a clean data set with bounded empirical covariance, resulting in much slower overall running time. For similar accuracy, the iterated filter algorithm with QUE scoring runs an order of magnitude faster than with naive spectral scoring.

**Further Remarks**    We thank the reviewers for pointing out several typos – we will fix them. We agree with Reviewer 1 about the missing centering step, which we will fix. We will clarify the definitions of $S_b, S_r$ and add a table of notation as suggested by Reviewer 3. We also agree about the error in Lemma 6.1, and with the suggested fix, which we will implement. Regarding Reviewer 3's question about equation (21): this was a typo (we thank you for pointing it out) – the correct step here is to bound the entire first term in the line above (b) by $\varepsilon\|M(w_t) - I\|\|\rho\|^2$. We have cleaned up this proof for future versions of the paper. If accepted, will increase the size of our plots using the additional page allowed in the camera-ready version.

*Figure:* We plot runtime vs mean estimation error (Euclidean norm) as fraction of points removed per iteration varies. We used synthetic data in 512 dimensions with $20\%$ outliers spread across 20 orthogonal directions. Each data point represents a single run of iterated filtering. Removing fewer points per iteration leads to more iterations and thus slower running times; these are the runs with high running time but (slightly) better accuracy.

[Meta-Review · NeurIPS 2019]

This is is a very strong paper which answers both theoretical and practical questions of robust estimation. It checks all the boxes since source code is also made available for reproducibility. Definitely one of the three best papers in my stack. I propose it for a spotlight presentation.